# Reactive wetting enabled anchoring of non-wettable iron oxide in liquid metal for miniature soft robot

Yifeng Shen [1], Dongdong Jin [1] ✉, Mingming Fu[1], Sanhu Liu[2,3], Zhiwu Xu[2,3], Qinghua Cao[4], Bo Wang[4], Guoqiang Li[1], Wenjun Chen[1], Shaoqin Liu[5] & Xing Ma [1,3,5] ✉

Magnetic liquid metal (LM) soft robots attract considerable attentions because of distinctive immiscibility, deformability and maneuverability. However, conventional LM composites relying on alloying between LM and metallic magnetic powders suffer from diminished magnetism over time and potential safety risk upon leakage of metallic components. Herein, we report a strategy to composite inert and biocompatible iron oxide ($Fe_3O_4$) magnetic nanoparticles into eutectic gallium indium LM via reactive wetting mechanism. To address the intrinsic interfacial non-wettability between $Fe_3O_4$ and LM, a silver intermediate layer was introduced to fuse with indium component into $Ag_xIn_y$ intermetallic compounds, facilitating the anchoring of $Fe_3O_4$ nanoparticles inside LM with improved magnetic stability. Subsequently, a miniature soft robot was constructed to perform various controllable deformation and locomotion behaviors under actuation of external magnetic field. Finally, practical feasibility of applying LM soft robot in an ex vivo porcine stomach was validated under in-situ monitoring by endoscope and X-ray imaging.

Soft robots are able to deform with a larger degree of freedom and absorb more energy arising from a collision in comparison to their rigid counterparts, which offer an opportunity to bridge the gap between machines and people[1,2]. When soft machines are further downscaled to small scale (millimeter and below), the derived miniature soft robots are expected to perform tasks in hard-to-access regions inside human bodies, and thus they are capable of revolutionizing extensive biomedical fields[3–8]. Various strategies have been developed to actuate and control soft robots at small scale, among which, magnetic miniature soft robots have particularly attracted widespread attention, thanks to the tissue transparency, remote operation capability, as well as accurate, rapid yet simple modulation of magnetic field[9–11]. Such types of robots are generally designed and

constructed by encoding magnetic agents (e.g., iron powder, neodymium iron boron microparticle, iron oxide magnetic nanoparticle, etc.) in soft material-based matrix, including elastomer, hydrogel and fluid[12–16], which allow to perform on-demand deformation and a range of locomotion behaviors under actuation of external magnetic field, such as rolling, crawling, jumping, swimming, and so on[17–21]. Specifically, compared to polymer-based machines, miniature soft robots constituted by magnetically responsive fluid generally possess a higher degree of softness and deformation freedom, making themselves more adaptative to dynamic and unstructured working environments for biomedical applications[22–25].

To date, water and organic solvents have been widely employed as the matrix of liquid-based soft robot, which, however, pose challenges

[1]Sauvage Laboratory for Smart Materials, School of Materials Science and Engineering, Harbin Institute of Technology (Shenzhen), Shenzhen 518055, China. [2]School of Materials Science and Engineering, Harbin Institute of Technology, Harbin 150001, China. [3]State Key Laboratory of Advanced Welding and Joining, Harbin Institute of Technology, Harbin 150001, China. [4]School of Materials Engineering, Shanghai University of Engineering Science, Shanghai 201620, China. [5]Key Laboratory of Microsystems and Microstructures Manufacturing, School of Medicine and Health, Harbin Institute of Technology, Harbin 150080, China. ✉ e-mail: jindongdong@hit.edu.cn; maxing@hit.edu.cn

in complex physiological conditions, due to their potential risks of adherence, evaporation, and toxicity[16,26]. In this regard, gallium-based liquid metal (LM) with appealing immiscibility, flexibility and biocompatibility provides an alternative[27–30]. Through amalgamating with metallic magnetic powders, e.g., Fe, Ni, NdFeB, and Gd[31–33], magnetic LM composites have been prepared, which can exhibit a variety of controllable robotic behaviors by programming magnetic fields[34–39]. Nevertheless, the inevitable alloying reaction between LM and metallic dopants occurs over time, which will induce transition in the crystal structures of magnetic materials, and thus gradually deteriorate the magnetically responsive performance[40]. In this regard, Lu et al. coated a silver shell on the surface of iron particle before mixing with LM, which could serve as a sacrificial layer to react with LM and thus protect iron particle from corrosion[40,41]. Nevertheless, the mixed magnetic powders may cause damage to living organisms if leaking or dissolving into surrounding environments, which is highly possible in current magnetic LM composites[42]. Therefore, emerging research focus is shifted to the composition of inert and biocompatible iron oxide ($Fe_3O_4$) magnetic particles into LM, while the most important issue is to address interfacial non-wettability between $Fe_3O_4$ and LM caused by the substantial mismatch in surface energy[43]. Although vigorous mechanical grinding has been reported to effectively facilitate inorganic oxides (e.g., graphene oxide, $Al_2O_3$, etc.) to composite with LM via coordination binding[44], it is still a time-consuming and labor-intensive procedure that requires the production of LM surface oxide layers to wrap on doping particles[45]. Consequently, such oxide films will attenuate the fluidity of LM, yet if they are eliminated, oxide particles will easily leak out under external stimulus to invalidate the prepared functional composite[46,47]. Therefore, an effective composition strategy between iron oxide and LM to guarantee satisfactory softness, magnetism and stability is highly desired for LM-based magnetic miniature soft robots.

Herein, we report a facile preparation method of magnetic LM composite that is capable of incorporating non-wettable iron oxide magnetic nanoparticles within eutectic gallium indium (EGaIn) via reactive wetting between LM and Ag shell modified $Fe_3O_4$ nanoparticle (Fig. 1). Through functionalizing $Fe_3O_4$ nanoparticle with polydopamine (PDA) layer and Ag nanoparticles, the interfacial wettability between magnetic agent ($Fe_3O_4$@PDA@Ag, FPA) and LM is significantly improved, allowing a rapid and thorough composition using diverse methods, including mechanical grinding, electrochemical fusion, and acid-facilitated amalgamation. Detailed microscopic characterization results indicate the $Ag_xIn_y$ intermetallic compounds (IMCs) formed by the alloying reaction between Ag and indium (In)

components serve as the anchoring sites to embed and fix $Fe_3O_4$ nanoparticles into EGaIn matrix, thus contributing to excellent suspension stability. In this manner, we succeed in preparing magnetic LM composites with desired fluidity and magnetism by elaborately regulating mass fraction of FPA, which can be employed as small-scale soft robot to perform on-demand deformation and locomotion behaviors under the actuation of external magnetic field. Furthermore, the developed soft robot that is immiscible with biological tissues is capable of dexterous navigation and targeted cargo transportation in an ex vivo porcine stomach under the control of a robotic arm-integrated magnet and in-situ monitoring by imaging modalities (e.g., endoscope and X-ray imaging). Therefore, our approach offers a generalized strategy to extend the applications of LM-based magnetic miniature soft robot for interventional therapy and minimally invasive surgery.

## Results
### Design and synthesis of iron oxide ($Fe_3O_4$)-based magnetic agent to regulate its wettability with LM

To be capable of compositing with LM, the component and surface properties of iron oxide ($Fe_3O_4$) magnetic nanoparticles were deliberately designed. Due to the wetting capacity with LM[48], silver was selected as an intermediate layer to functionalize $Fe_3O_4$ nanoparticles, resulting in a core-shell structured magnetic agent. Figure 2A presented the preparation process, in which, monodispersed $Fe_3O_4$ nanoparticles with an average diameter of ~850 nm (Fig. S1) were synthesized by hydrothermal method[49]. Then, a polydopamine layer that could effectively chelate with heavy metal ions[50] was used to completely cover the surface of $Fe_3O_4$ nanoparticles ($Fe_3O_4$@PDA) through the self-polymerization reaction of dopamine in weak alkaline buffer[51]. The average size of fabricated $Fe_3O_4$@PDA powders was $960 \pm 120$ nm (Fig. S2). Finally, with the assistance of dextroglucose reductants, the Ag ions chelated onto the PDA layer in advance were in situ reduced via silver mirror reaction, which immobilized a large number of Ag nanoparticles on the surface of $Fe_3O_4$@PDA (Fig. S3), contributing to the preparation of $Fe_3O_4$@PDA@Ag (FPA) nanoparticles.

Subsequently, Fig. 2B displayed the transmission electron microscopy (TEM) image and energy dispersive X-ray (EDX) spectroscopy of obtained magnetic agent, which clearly demonstrated the core-shell structured morphology, as well as the elemental distribution of Fe, O, N, and Ag. Fourier-transform infrared (FT-IR, Fig. 2C) spectroscopy and X-ray diffraction (XRD, Fig. 2D) patterns were used to characterize the change of components during preparation. Compared to pristine $Fe_3O_4$ nanoparticles, new peaks appeared at 2917, 2850, 1490, and

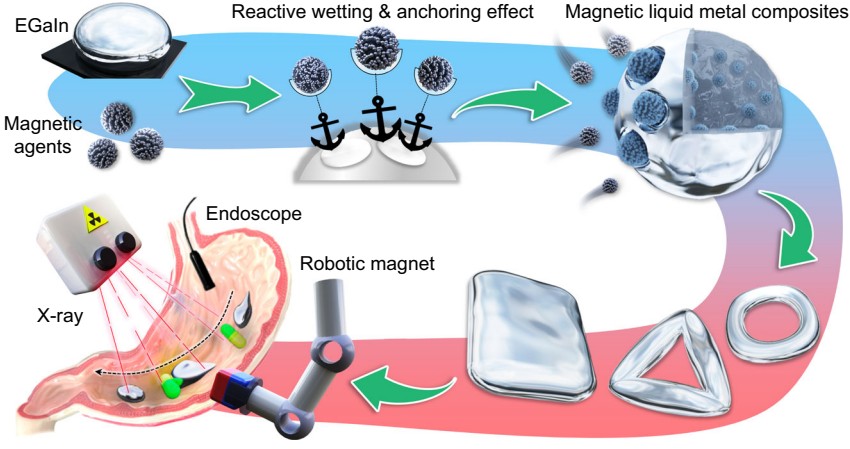

**Fig. 1 | Scheme of the preparation of liquid metal-based magnetic miniature soft robot via reactive wetting and anchoring effect.** The targeted delivery application in stomach under the guidance of endoscope and X-ray imaging system is also exhibited.

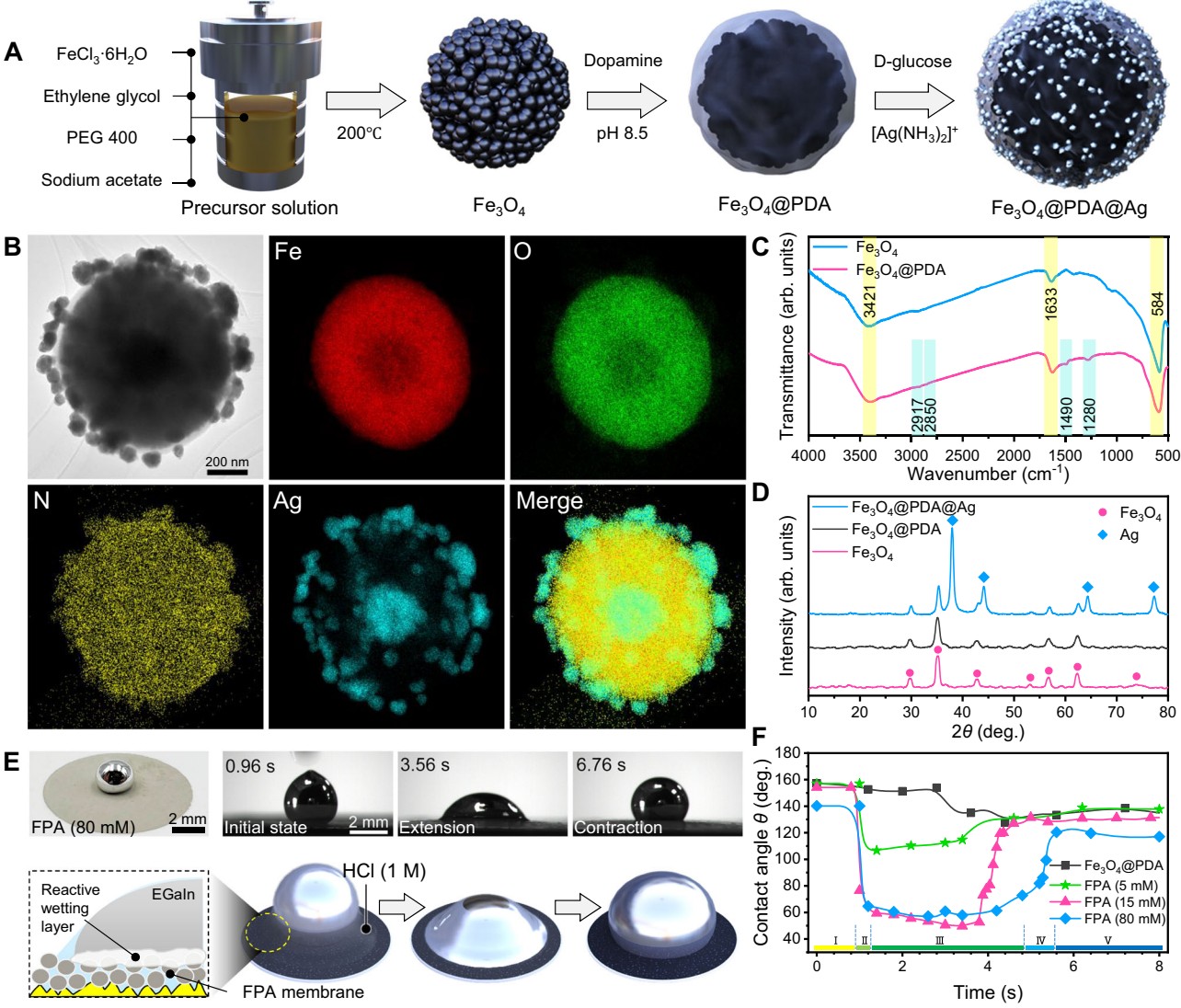

**Fig. 2 | Preparation and wettability regulation of iron oxide magnetic agent.**
**A** Schematic of the preparation process of Fe₃O₄@PDA@Ag (FPA) particles.
**B** Transmission electron microscope (TEM) images of the core-shell structured FPA (5 mM) nanoparticle and the corresponding element mappings. Each experiment was repeated independently for 3 times with similar results. **C** Fourier-transform infrared (FT-IR) spectroscopy of Fe₃O₄ and Fe₃O₄@PDA. The yellow region indicated the common peaks for two kinds of particles, while the light blue region suggested the characterization peaks that only Fe₃O₄@PDA possessed. **D** X-ray diffraction (XRD) patterns of Fe₃O₄, Fe₃O₄@PDA, and FPA (5 mM). **E** Snap images and illustrations of the EGaIn droplet extension and contraction on the FPA (80 mM) membrane showing the wettability transition triggered by the alloying reaction between Ag and EGaIn. **F** Variation of contact angle as a function of time during the wettability tests on Fe₃O₄@PDA, FPA (5 mM), FPA (15 mM), and FPA (80 mM) membranes, where the whole dynamic process is divided into five stages. Source data are provided as a Source Data file.

1280 cm$^{-1}$ for Fe₃O₄@PDA nanoparticles, suggesting the existence of C-H bonds and aromatic rings from PDA layers[52]. While for the XRD results, although the PDA coating process did not induce an obvious change in the patterns of Fe₃O₄, distinct characteristic peaks of Ag crystal were detected after the silver mirror reaction, which were consistent with the selected area electron diffraction result in Fig. S4. It was worth noting that by changing the concentration of Ag$^+$ (5–100 mM) during the silver mirror reaction, the amount of Ag nanoparticles loaded on FPA could be adjusted, as shown in Fig. S5. Finally, an intact and thick Ag shell could be obtained when the concentration of Ag$^+$ was > 40 mM. In addition, the magnetic properties of FPA nanoparticles were evaluated in Fig. S6, demonstrating a paramagnetic characteristic whose saturation magnetization would decrease with the elevation of the loading amount of diamagnetic silver nanoparticles. Therefore, all the above results verified the successful preparation of magnetic agent FPA, which consisted of a Fe₃O₄

nanoparticle core, a PDA middle layer, and abundant Ag nanoparticles decorated on the outer surface.

With the loaded Ag nanoparticles as wetting sites, FPA nanoparticles should become possible to composite with LM. To verify this hypothesis, we measured the contact angle between FPA and LM by adding EGaIn droplet (~10 μL to avoid the influence of gravity) to a membrane substrate that filtered FPA nanoparticle solution in advance. After filtering a range of FPA nanoparticles with the increased loading amount of Ag nanoparticles, the color of membrane substrates would change from black to silvery white (Fig. S7). Besides, before measuring the contact angle, a few hydrochloric acid solution (1 M, ~5 μL) was dropped on EGaIn droplet to remove the gallium oxide skin of LM, and thus exhibited the real wettability relationship between FPA and EGaIn. As shown in Fig. 2E, when FPA (80 mM) membrane was used, the peach-shaped EGaIn droplet (contact angle was above 140°) immediately became flattened with a contact angle of <59.8° upon

adding HCl, indicating the good wettability between FPA (80 mM) and EGaIn. However, as time went on, the EGaIn droplet recoiled to a spherical shape, associated with the increase of contact angle to 118.6°. Quantitative variation in the contact angle of EGaIn droplet on different FPA membranes with a function of time was presented in Fig. 2F, which could be further divided into five stages (more detailed information in Fig. S8 and Movie S1). Initially, EGaIn droplet did not spread on any substrates (stage I) due to the presence of EGaIn oxide film. The specific contact angles were determined by the shear stress that EGaIn droplet encountered when being squeezed out from a needle, the viscoplastic property, and the surface tension of oxided liquid metal, rather than the real wettability relationship between EGaIn and FPA[53,54], as shown in Fig. S9. Upon the addition of HCl (stage II), the contact angles between EGaIn droplet and different FPA membranes decreased rapidly (50.4°, 77.6°, and 71.3° drop in < 40 ms for FPA (5, 15, 80 mM), respectively), and then reached a metastable condition called stage III. At this stage, the EGaIn droplet could spread out on FPA (>15 mM) membrane, in which, their average contact angles were <70° (Fig. S10). The metastable condition could be maintained for several seconds, followed by the recovery of contact angle at stage IV. Finally, the geometry of EGaIn droplet returned to sphere (stage V), whose average contact angle was 15.4% lower than that at the first stage when the substrate was FPA (80 mM). Such special and dynamic process mainly included two stages, i.e., the extension and contraction of LM droplet, which was obviously different from the wetting behavior between EGaIn and bulk silver[55]. It was worth noting that for Fe$_3$O$_4$@PDA nanoparticles, only a slight variation in contact angle was observed, indicating the indispensable role of Ag in regulating the wettability between magnetic agents and LMs (Fig. S11).

To figure out the reason why the dynamic phenomenon happened, the micromorphology and component of the FPA (100 mM) filter membrane after contact angle test were investigated. From the top view shown in Fig. S12, the porous structure and nanoparticle morphology were clearly observed for the pristine filter membrane and magnetic agents, respectively. At the same time, the LM was found to spread out on the FPA (100 mM) nanoparticle layer at the boundary between EGaIn and FPA, which agreed well with the macroscopic wetting results. Besides, Fig. S13 provided the component analysis from a cross-section view, in which gallium element could only be detected in LM area while In element generally distributed in both EGaIn and FPA areas. This result indicated an underlying reaction between the In component of EGaIn and FPA nanoparticles, which should constitute the reactive wetting mechanism between LM and magnetic agents. Meanwhile, the phagocytosis effect[55] and the tensile force[56] of EGaIn eroded the coated Ag particles over time, exposing the Fe$_3$O$_4$@PDA surface to EGaIn, leading to the deterioration of wettability and the breaking of metastability. As a result, the dynamic phenomenon accompanied a morphological reconfiguration process of EGaIn droplet, which was obtained via wettability transition without external power, in contrast to the wetting-dewetting behavior of LM droplets stimulated by electric field[57,58]. More importantly, the wettability between iron oxide and EGaIn was significantly improved when Fe$_3$O$_4$ nanoparticles were coated by enough Ag nanoparticles, paving the foundation of preparing magnetic LM composites.

## Anchoring effect of magnetic agent in LM via reactive wetting

After modifying the component and surface properties of Fe$_3$O$_4$ nanoparticles with silver, the composition between the obtained magnetic agent and LM was investigated. In consideration of better wettability, we first mixed FPA (100 mM) with EGaIn to fabricate the magnetic LM composites. Thanks to the acceptable wettability improved by silver coating, several approaches could be employed to prepare EGaIn-FPA (100 mM), including mechanical grinding, electrochemical amalgamation, and acid-facilitated amalgamation. As shown in Fig. 3A and Movie S2, in sharp contrast to the immiscibility between

EGaIn and pristine Fe$_3$O$_4$ powders, a stable and glossy LM-based composite was obtained after combining FPA (100 mM) nanoparticles with EGaIn for several seconds, which could be attributed to the reactive wetting mechanism as explained in previous section. For mechanical grinding, cracking the surface oxide film in the grinding process could stimulate the direct reaction between interior EGaIn and the Ag shells of FPA (100 mM), thus accelerating the combination (~10-fold faster than that between EGaIn and Fe$_3$O$_4$). Such result was consistent in oxygen-free condition (Fig. S14), where FPA (100 mM) nanoparticles were able to compound with EGaIn in a glovebox filled with nitrogen gas via slight grinding, but bare Fe$_3$O$_4$ powders could not. While for the latter two methods (Movie S3), external electrical fields and strong acids were employed to eliminate the oxide skins via electrochemical reduction and acid corrosion, respectively[57], which also realized the direct contact and wetting between exposed EGaIn and FPA (100 mM).

The suspension stability of magnetic agent inside liquid medium is crucial for sustainable and reliable applications, as the inner magnetic particles may be separated out by the imposed magnetic force during actuation, which would lead to the attenuation of magnetic responsiveness. Therefore, the magnetic particle leakage from magnetic liquid metal composite was first quantified by immersing composites into acidic solution and evaluating the environmental Fe element concentration after applying a magnetic field (~200 mT) for 24 h (Fig. 3B and Fig. S15). At this time, the content of Fe$_3$O$_4$ was controlled to be 1.25 wt% in both EGaIn-FPA (100 mM) and EGaIn-Fe$_3$O$_4$ groups in advance. It could be found that the Fe element concentration of the former group (6.62 mg/L) was significantly lower than that of the latter one (29.16 mg/L), indicating the better suspension stability of FPA (100 mM) in EGaIn matrix. The interior microstructure of EGaIn-FPA (100 mM) with an FPA mass fraction of 5% was further investigated by freezing the composite and checking sectional constituents using a metallographic method. As shown in Fig. 3C, a phase separation phenomenon similar to the previous work occurred[48], in which the profile of Ga element was complementary with In element. The black dots representing Fe$_3$O$_4$ nanoparticles were found to only distribute in the In element region, whose density would increase with the elevation of FPA mass ratio as shown in Fig. S16. Meanwhile, it could be found that the distribution of Ag element exceeded the area of Fe element and appeared to highly overlap with In element region, verifying that the reaction between the Ag shell of FPA nanoparticles and the In phase of EGaIn LM facilitated the blend of iron oxide magnetic nanoparticles.

To detect and infer the reactive wetting product, XRD analysis was conducted for the cross-sectional metallographic structures of EGaIn-FPA composites with different Ag loading amounts. As shown in Fig. 3D, when FPA (20 mM) nanoparticles was blended into EGaIn with a mass fraction of 10%, AgIn$_2$ IMCs were formed according to the XRD characteristic peaks. If FPA (100 mM) nanoparticles were employed instead, the IMCs became Ag$_9$In$_4$, indicating the increase of Ag dopant would alter the phase structure of IMC, which was consistent with the Ag-In phase diagram[59]. Therefore, in spite of varied crystal structure, the presence of intermetallic phase with metallic bond interaction[60] not only facilitated the wetting of FPA with EGaIn, but also anchored the magnetic agents in LM matrix, thus contributing to the better suspension stability than that of bare Fe$_3$O$_4$ in EGaIn. Moreover, the in situ three-dimensional distribution of IMC inside EGaIn-5% FPA (100 mM) composite was explored using the micro-computed tomography technique (micro-CT). As shown in Fig. 3E and Movie S4, the high-density Ag$_x$In$_y$ phase (green region) distributed uniformly in the EGaIn matrix (the region surrounded by dotted boundary). Through further zooming in a profile of the stereogram, low-density Fe$_3$O$_4$ nanoparticles were found to anchor steadily in the EGaIn via the restriction of Ag$_x$In$_y$ IMC, which agreed well with the metallographic analysis. Besides, it was worth noting that the irregular shape of IMC would hinder the flow of interior LM and increase the local viscosity

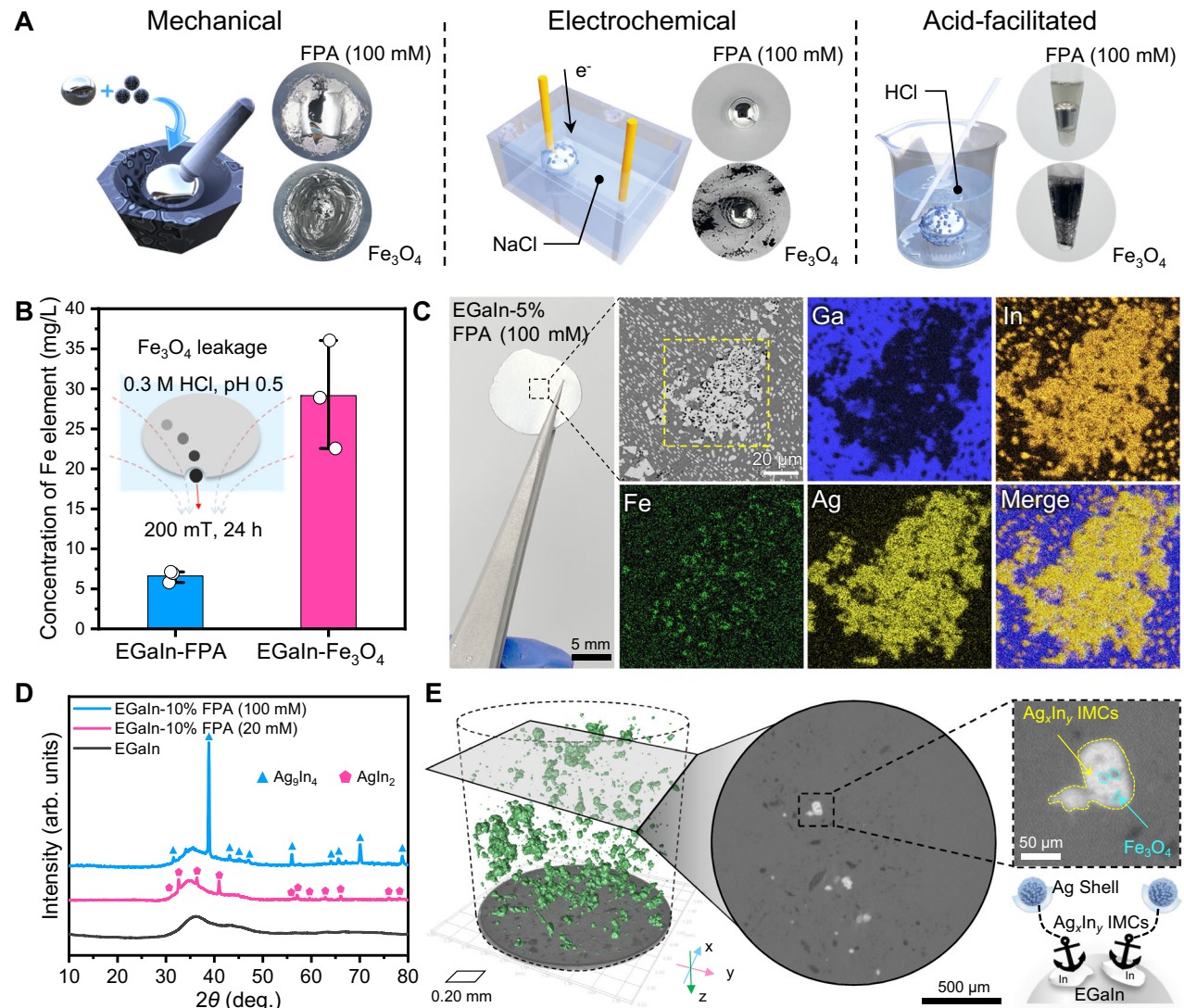

**Fig. 3 | Preparation of magnetic liquid metal composite and mechanism of anchoring effect. A** Illustrations and optical images of the composition between EGaIn liquid metal and FPA (100 mM) and $Fe_3O_4$ nanoparticles using mechanical, electrochemical, and acid-facilitated methods, respectively, indicating the superior wettability of FPA (100 mM) in LM. **B** Concentrations of Fe element measured from the supernatant of acid solution containing EGaIn-5% FPA (100 mM) and EGaIn-1.25% $Fe_3O_4$, respectively. All values represent the mean ± standard deviation (SD) for $n = 3$ independent experiments. **C** Metallography characterization of the frozen EGaIn-5% FPA (100 mM) where the yellow dotted region is further analyzed by

element mapping. **D** XRD patterns of EGaIn-10% FPA (100 mM), EGaIn-10% FPA (20 mM), and pure EGaIn. **E** Micro-computed tomography (Micro-CT) of EGaIn-5% FPA (100 mM), where the 3D image (left) shows the distribution of high-density intermetallic compound phase (green regions) in liquid metal (cylindrical dotted region), the slice image (middle) exhibits the horizontal section of EGaIn-5% FPA (100 mM) composite, and the illustration in the right corner indicates the anchoring mechanism between $Fe_3O_4$ and EGaIn via $Ag_xIn_y$ intermetallic compounds (IMCs). Source data are provided as a Source Data file.

around the embedded magnetic agents[61], leading to the apparent reduction of FPA leakage and the stability improvement of LM-based composite no matter in solid or fluid condition.

Furthermore, we noted that only $Ag_xIn_y$ IMCs rather than $Ag_xGa_y$ IMCs were detected in the EGaIn-FPA composite. The surface indium enrichment effect of EGaIn LM might facilitate the preferential formation of $Ag_xIn_y$ phase due to the priority exposure to FPA[62–64]. But the extremely thin enrichment layer (3-4 atomic diameters) could hardly exhaust the silver atoms on the surface of FPA immediately upon mechanical grinding. As a result, direct contact between Ga, In and Ag inevitably occurred, and we attributed the formation of $Ag_xIn_y$ IMCs to the reaction preference between In and Ag. To support the hypothesis, we conducted several control experiments. In the first one, we mixed FPA (100 mM) particles with pure Ga LM via mechanical grinding, and the corresponding XRD characterization was presented in Fig. S17. The presence of prominent $Ag_2Ga$ peaks indicated that Ag shell of FPA

could indeed react with gallium to produce $Ag_xGa_y$ IMCs in the absence of indium[65]. Then, in the second control experiment, we ground Ag particles with EGaIn LM. The XRD results in Fig. S18 indicated that with the increase of Ag particle amount, the formed IMC phase in LM composite gradually changed from $AgIn_2$ to $Ag_9In_4$. While $Ag_xGa_y$ IMC did not occur even when the mass ratio between Ag and EGaIn was as high as 50% (i.e., mixing 10 g EGaIn with 5 g Ag particle). In the third control experiment, 0.5 g Ag particle was first mixed with 10 g pure Ga LM, followed by the addition of 1 g In powder. In Fig. S19, $Ag_2Ga$ phase was initially identified in Ga-Ag mixture by XRD characteristic peaks. However, $Ag_2Ga$ peaks disappeared after In powder was added, while new peaks corresponding to $Ag_9In_4$ phase appeared instead, indicating Ag atoms were more inclined to bind with In atoms compared to Ga atoms. From the thermodynamic viewpoint of In-Ag and Ga-Ag systems, it was reported that the enthalpy of mixing in In-Ag alloy was generally lower than that in the Ga-Ag system[66,67], which

indicated the interactions between Ag and In atoms were stronger than those between Ag and Ga atoms. Therefore, in the Ga-In-Ag ternary system, reaction preference between In and Ag existed, which could be the reason why only $Ag_xIn_y$ IMCs were detected when we mixed EGaIn LM with FPA particles. During the composition between FPA and EGaIn, the silver modification enabled the incorporation and anchoring of non-wettable iron oxide in LM via reactive wetting mechanism, promoting the next-step development of biocompatible magnetic LM soft robot.

## On-demand deformation and manipulation of magnetic LM composite for miniature soft robot

After incorporating iron oxide magnetic agents, the obtained magnetic liquid metal composites became able to behave in response to external magnetic field, thus emerging as a promising candidate for soft robotic applications. The rheological property and magnetism of liquid metal magnetic soft robot (LMMSR), which played a vital role in tasking performance, were first investigated. As shown in Fig. 4A, when elevating the mass fraction of embedded FPA (100 mM) nanoparticles from 0 to 10%, the viscosity of EGaIn-FPA (100 mM) composite increased gradually with the texture transforming from liquid into slurry. Upon encountering shear stress, the nanoparticles inside EGaIn by anchoring effect hindered the liquid matrix from flowing freely, leading to the viscosity being positively correlated with mass fraction of the magnetic particle. However, EGaIn-5% FPA (100 mM) still maintained an excellent fluidity closed to pure LM. Besides, similar to pure LM[68], EGaIn-FPA (100 mM) composite also exhibited shear-thinning properties, i.e., the higher the shear rate, the lower the viscosity, which could be attributed to the broken of the inner net structures formed by dense magnetic particles during rheological shearing. Moreover, under a constant low strain amplitude (5%), the storage modulus and the complex viscosity of EGaIn-FPA (100 mM) gradually increased when enhancing magnetic particle fraction, as presented in Fig. 4B. In addition, as shown in Fig. S20, different preparation methods might induce a slight difference in the rheological properties of magnetic LM composites, and herein all the following LMMSRs were prepared by mechanical grinding if not specifically stated. The durability of EGaIn-5% FPA (100 mM) LM composite in terms of rheological property was also provided, which indicated the storage in 1 week did not significantly affect the fluidity of soft robot. Except for rheological properties, Fig. S21 characterized the magnetic property of LMMSR. It could be found that the paramagnetic feature of FPA (100 mM) did not alter after compounding with EGaIn, and endowed the composite with a saturation magnetization of 3.5 emu/g when the particle weight percentage was 20%, making it possible to actuate and control a LMMSR using an external magnetic field.

Afterward, the deformation capability of EGaIn-FPA (100 mM) composite-based LMMSR was studied. Two deformation models existed upon the application of external magnetic field, including passive and active transformation. The former referred to the behavior that a LMMSR was pressed to change its fluid shape due to the restriction of surrounding boundaries during actuation. In comparison, the latter represented the spontaneous deformation behavior when the LMMSR encountered a magnetic field with nonuniform strength distribution. For example, Fig. 4C demonstrated that the LMMSR would become flattened when a permanent magnet was positioned below its substrate, leading to a decrease in aspect ratio of LMMSR (height/width). At this time, as the FPA (100 mM) magnetic nanoparticles anchored in EGaIn matrix were subjected to a static gradient magnetic field, magnetic force $\mathbf{F}_m = V_m(\mathbf{M} \cdot \nabla)\mathbf{B}_{ext}$ would be generated, which pressed the LMMSR onto the substrate and thus transformed its shape from ellipsoidal to pie-shaped. In the equation, $V_m$ was the volume of magnetic particles, $\mathbf{M}$ was the total magnetic dipole moment, and $\mathbf{B}_{ext}$ represented the external magnetic field. Consequently, more magnetic agents embedded in the

composite and stronger external magnetic field resulted in the larger deformation of LMMSR when the viscosity was not significantly changed, which was reflected by the lower aspect ratio value in Fig. 4C. Besides, it was worth noting that there existed an oxide skin with high surface stress (~0.5 N/m[69]) outside LM in oxygenous neutral environment, which would impair the deformation performance of LMMSR. Therefore, to maintain the satisfactory fluidic property, the LMMSR was immersed in acidic HCl solution (Fig. S22) for the following experiments.

The passive deformation performance was illustrated and quantified in Fig. 4D by dragging a LMMSR to traverse a narrow channel via a permanent magnet. When the mass ratio of magnetic agent was large enough or the distance between LMMSR and magnet was close enough, the induced magnetic force would be strong enough to push the LMMSR, and as a result, the shape of LMMSR was squeezed by the boundaries of channel. As shown in the phase diagram, it was found that the LMMSR was even able to pass through a slit whose width was 70% smaller than that of LMMSR (zone 1) via such passive deformation. With the dilution of FPA and attenuation of external magnetic field, the deformation capability of LMMSR gradually deteriorated (e.g., zone 2 with the maximum deformation rate of 40%), and finally made itself fail to navigate across the channel as shown in zone 3. In the following experiments, we fixed the concentration of FPA nanoparticles as 5% to optimize the rheological property and magnetic responsiveness. Besides, through changing the gradient distribution of magnetic field, different active transformation patterns could be reconfigured in a LMMSR. As shown in Fig. 4E and Movie S5, we used three types of permanent magnets (rectangular, annular, and triangle shapes) to spatially adjust the actuating magnetic field, under which, the imposed magnetic force drove the embedded magnetic agents to the location with maximum magnetic flux density, and thus transformed the LM into corresponding geometries. By altering the external magnets, LMMSR could switch between different shapes within seconds, demonstrating the agile and rapid active deformation capability. Furthermore, the stability in magnetic actuation performance of LMMSR over actuation cycles and storage time was systematically investigated. Detailed methods and results could be found in the Experimental Section and Fig. S23, 24, respectively. Repeated actuation under external magnetic field only made a little difference in the actuation performance of LMMSR due to the limited leakage of magnetic agents from composite (Fig. S25, 26), while the storage time did not affect the deformation ability of LMMSR in 1 week. These results demonstrated that our LMMSR prepared via reactive wetting mechanism possessed good stability.

Except for deformation performance, a LMMSR was also required to conduct tasks on demand, such as obstacle surmounting, cargo transportation, splitting, merging, and so on, aiming to satisfy the demands of versatile application scenarios. For example, as shown in Fig. 5A and Movie S6, through moving an external magnet along the trajectories of L and M, the LMMSR in acid solution (1 M HCl) could accurately follow the path. Figure 5B presented that in an S-shaped channel, two LMMSRs on opposite ends were successively steered by a magnet to traverse narrow paths and then encounter each other in the center ($t = 12$ s). Then, due to the fluidic nature, two LMMSRs would smoothly merge into a larger LMMSR that possessed all the properties that the former had. Therefore, after fusion, the LMMSR could still transform its shape to accommodate surrounding changing boundaries, which turned an orthogonal corner and finally arrived at the destination (Movie S7). More complex tasks including navigation, deformation, cargo transportation and splitting were accomplished by LMMSR in Fig. 5C and Movie S8. At first, the LMMSR with an ellipsoidal shape was placed at the starting point, and we used a rectangular magnet to navigate the LMMSR across a confined channel with zigzag corners via passive deformation ($t = 0$–28 s). When the soft robot

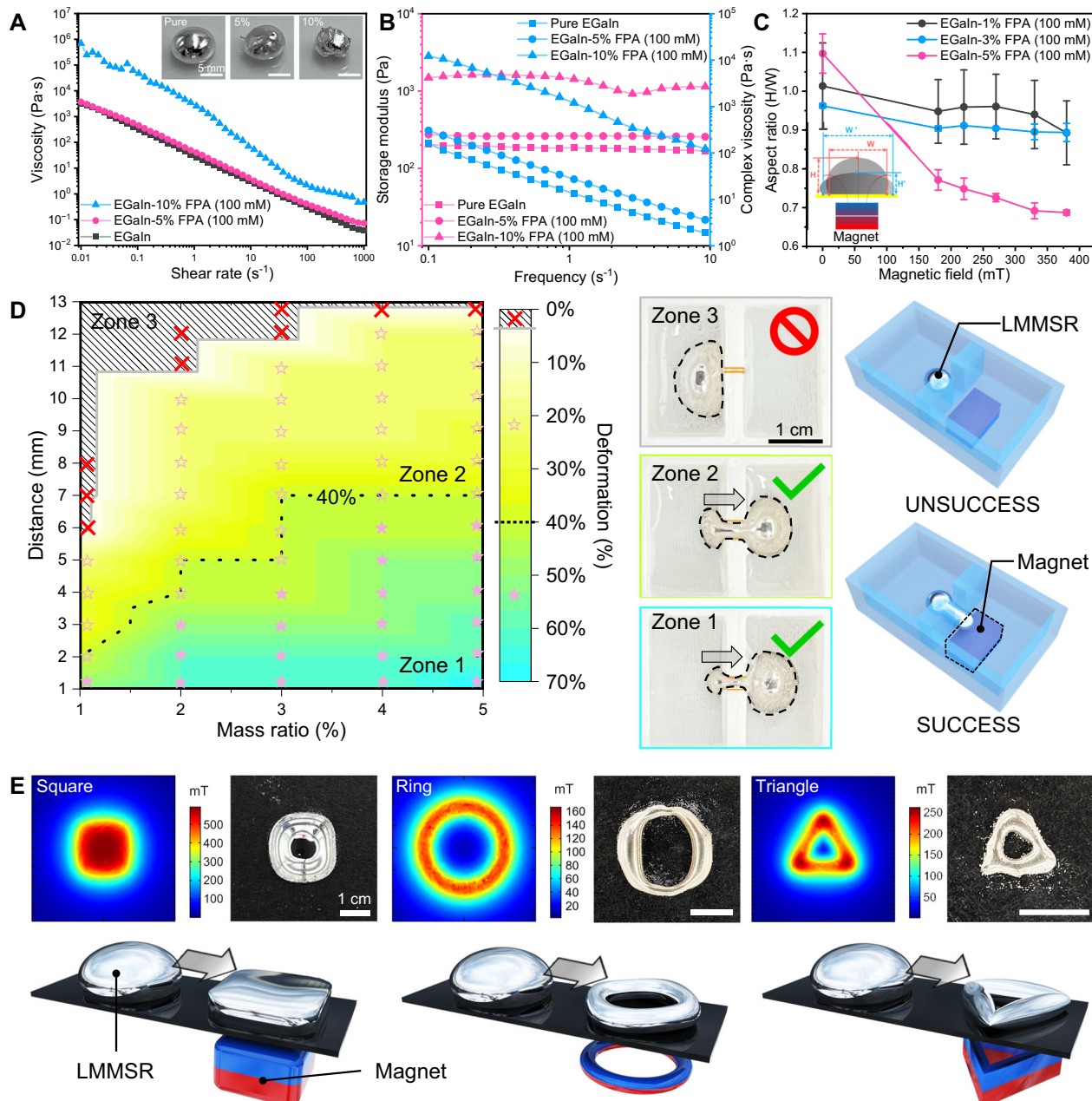

**Fig. 4 | Controllable deformation of liquid metal magnetic soft robot (LMMSR).**
**A** The viscosity of pure LM, EGaIn-5% FPA (100 mM) and EGaIn-10% FPA (100 mM). The insert optical photos show the appearance of liquid metal composites.
**B** Storage modulus and complex viscosity of pure LM and LMMSRs with different FPA (100 mM) mass ratios under various frequencies. **C** Aspect ratios measured from EGaIn-1% FPA (100 mM), EGaIn-3% FPA (100 mM), and EGaIn-5% FPA (100 mM) under different external magnetic fields. All values represent the mean ± SD for $n = 3$ independent experiments. **D** Phase diagram showing passive deformation capacities of LMMSRs with the variation of FPA (100 mM) mass ratio and distance between permanent magnet (NdFeB, N52, 24 mm by 24 mm by 12 mm) and LMMSR.

Zone 1 represents that the LMMSR can pass relatively narrow channels with a deformation degree of 40–70%, zone 2 represents that the LMMSR can pass relatively wide channels with a deformation degree < 40%, and the LMMSR can not pass channels in zone 3. The middle optical images and the right illustrations show the behaviors of LMMSRs in narrow slits with different widths. **E** Active deformation of LMMSRs under the actuation of permanent magnets with rectangular, annular, and triangle shapes, where the simulation results demonstrate the magnetic field distribution on the plane 1 mm above magnets. Source data are provided as a Source Data file.

reached the position with a capsule cargo, the position of magnet was carefully adjusted to make the LMMSR push the cargo into a predesigned holder ($t = 28–48$ s). Subsequently, a ring-shaped magnet replaced the former magnet and initiated the active deformation of LMMSR with the ring-distributed magnetic field flux. Through quickly changing the distance between LMMSR and magnet, the soft robot was stretched by magnetic force and started to split into two sub-LMMSRs ($t = 48–74$ s) once their diameter was larger than the critical

wavelength of Rosensweig pattern[23,70]. Finally, the formed two sub-LMMSRs with distinct sizes were navigated to their own destinations in sequence, indicating the accomplishment of tasks. Therefore, after anchoring iron oxide magnetic agents inside LM, the produced composite was endowed with excellent softness and magnetism, which allowed on-demand deformation and manipulation via external magnetic field, thus emerging as a promising soft robot for targeted delivery application.

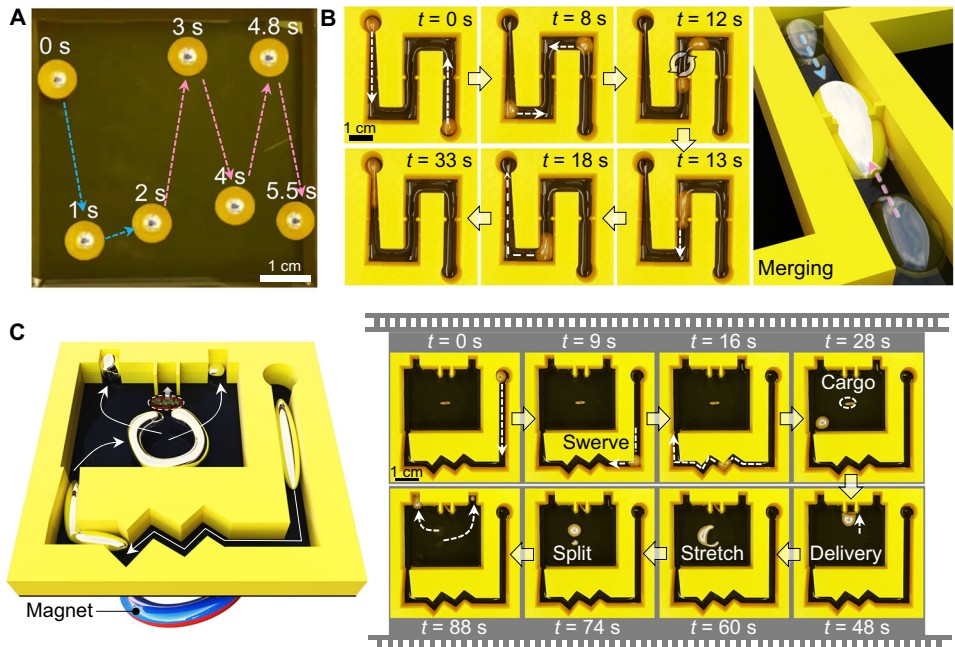

**Fig. 5 | On-demand magnetic manipulation of LMMSR. A** Overlapped sequential snapshots indicating the LMMSR is magnetically navigated along L and M trajectories in an open tank filled with HCl solution (1 M). **B** Image sequences and illustration showing the locomotion and fusion of LMMSRs in a channel. **C** Illustration and sequential video snapshots showing the deformation, cargo delivery, and splitting of LMMSR by programming external magnetic field.

## Targeted transportation in ex vivo porcine stomach under the guidance of imaging modalities

The cytotoxicity of EGaIn-FPA composite, which was crucial for biomedical applications, was first investigated before performing tasks. During the test, the LMMSR was broken up into microparticles (Fig. S27) by ultrasonic atomization to improve the interaction between composite and testing cells. The obtained particles were co-cultured with two normal cell lines, including human embryonic kidney cells (HEK293) and mouse embryonic fibroblasts (3T3). Figure 6A exhibited that the microparticles were endocytosed by HEK293 cell after co-incubation for 4 h, preliminarily suggesting that normal cells treated by our composite material remained cytoactive (detailed information was shown in Fig. S28). Next, a nonradioactive colorimetric Cell Counting Kit-8 (CCK-8) assay was carried out to quantitatively evaluate the viabilities of normal cells after being co-cultured with different concentrations of particles for 24 h. As shown in Fig. 6B, our magnetic LM composites were not harmful to both HEK293 and 3T3 cells when the concentration was < 50 μg/mL. The cell viability might experience a slight decrease when the microparticle amount was increased, which, however, still maintained above 80% with a LM composite concentration as high as 200 μg/mL. So, the developed magnetic LM composite exhibited satisfactory biocompatibility, which was better than that of many commercially available ferrofluids (Fig. S29). A live/dead cell staining experiment was further performed in Fig. S30, which showed that the majority of HEK293 and 3T3 cells were in a satisfying living state. Moreover, the leakage of metal elements (Ga, In, Fe and Ag) from EGaIn-FPA composite was measured after immersing 500 mg EGaIn-5% FPA in 1 mL simulated gastric acid (0.1 M HCl). The magnetic agents were found to be almost stable in such acidic condition (Fig. S31). While for the magnetic LM composite, as shown in Fig. 6C, negligible In and Ag ions were detected in the solution. The concentration of Ga and Fe slowly increased as time went on, which could be attributed to the dissolution of the oxide film of LMMSR and subsequent leakage of magnetic agents. After 1 h of incubation, the concentration of leaked Ga, In, Fe and Ag was 17.40, 1.90, 33.52, and 3.94 mg/L, respectively. Through reproducing the concentration of each metal element in cell culture medium, it was

found that these leaked metal ions showed very limited cytotoxicity to HEK293 cells (Fig. S32). Therefore, it could be concluded that our developed LMMSR exhibited excellent biocompatibility, which paved the foundation for next-step biomedical applications.

As an acidic environment was usually required to guarantee satisfactory rheological property of EGaIn-FPA composite, the application scenario of LMMSR focused on the biological environment inside stomach. To verify this goal, an ex vivo porcine stomach filled with simulated gastric acid was used. As shown in Fig. 6D and Fig. S33, a LMMSR was injected into stomach under the real-time monitoring of an industrial endoscope, aiming to transport a drug capsule stuck in the mucosa (Movie S9). The capsule was dyed in malachite green for easy visualization. Figure 6E showed that under the actuation of external magnetic field, the LMMSR strode over the acid-air interface and climbed along the gastric wall (stage i). Due to the strengthened anchoring effect between magnetic agent and LM, the LMMSR was able to overcome its gravity instead of bringing about particle separation. After arriving at the location where the capsule stuck, the LMMSR began to propel the capsule back to the gastric acid by manually moving the position of magnet (stage ii, t = 24–33 s). At the final stage, the capsule was pushed into acid solution, followed by the release of loaded drugs (yellow dyed), thus realizing the targeted cargo delivery in an ex vivo stomach. During the transportation process, magnetic LM composite exhibited appealing immiscibility with biological tissue, which was a notable advantage over many other conventional ferrofluids for biomedical applications (Fig. S34).

Taking advantage of the high mass density of EGaIn, X-ray-based imaging system was further employed to monitor the status of soft robot inside ex vivo porcine stomach. As shown in Fig. 7A and Fig. S35, after LMMSR injection, the stomach filled with simulated gastric acid was sealed with surgical suture to prevent acid leakage and then placed in the imaging system that mainly consisted of X-ray source and detector. A magnet fixed on robotic arm was mounted inside the system for magnetic control. Through adjusting the position and posture of robotic arm, the external magnetic field could be modulated, thus allowing to dexterously actuate the LMMSR in a remote manner (Fig. 7B and Movie S10). A camera was also positioned in the

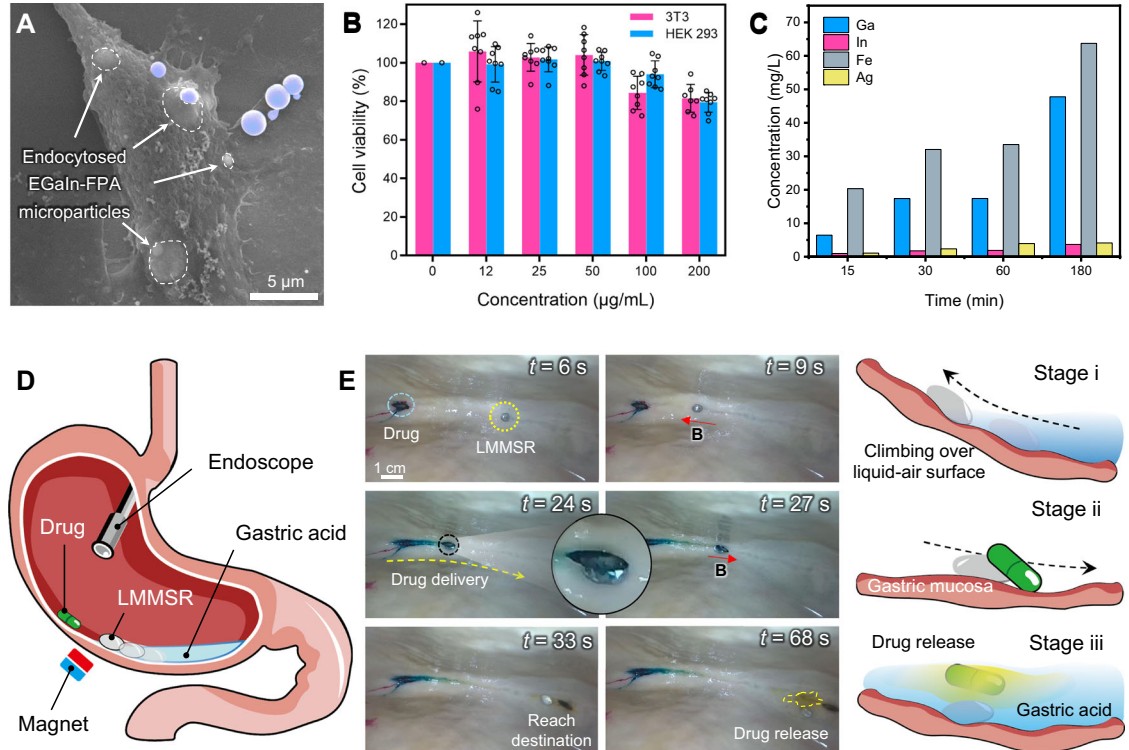

**Fig. 6 | Biocompatibility analysis of LMMSR and drug capsule delivery demonstration in an ex vivo porcine stomach. A** Scanning electron microscope (SEM) image showing the endocytosis of EGaIn-5% FPA (20 mM) microparticles by HEK293 cell, where purple areas represent the free liquid metal microparticles. **B** Cell viability of HEK293 and 3T3 cells cultured with EGaIn-5% FPA (20 mM) microparticles at various concentrations for 24 h. All values represent the mean ± SD for n = 8 independent experiments. **C** The concentrations of heavy metals measured from the supernatant of HCl solution (0.1 M) containing EGaIn-5% FPA (20 mM) as a function of time. The values were obtained by the average number of 2 independent experiments. **D** Illustration showing the magnetic manipulation of LMMSR in an ex vivo porcine stomach under the real-time guidance of endoscope. **E** Sequential snapshots and illustrations showing the drug capsule delivery process using LMMSR, which is divided into three stages, i.e., climbing, transporting, and drug release. Source data are provided as a Source Data file.

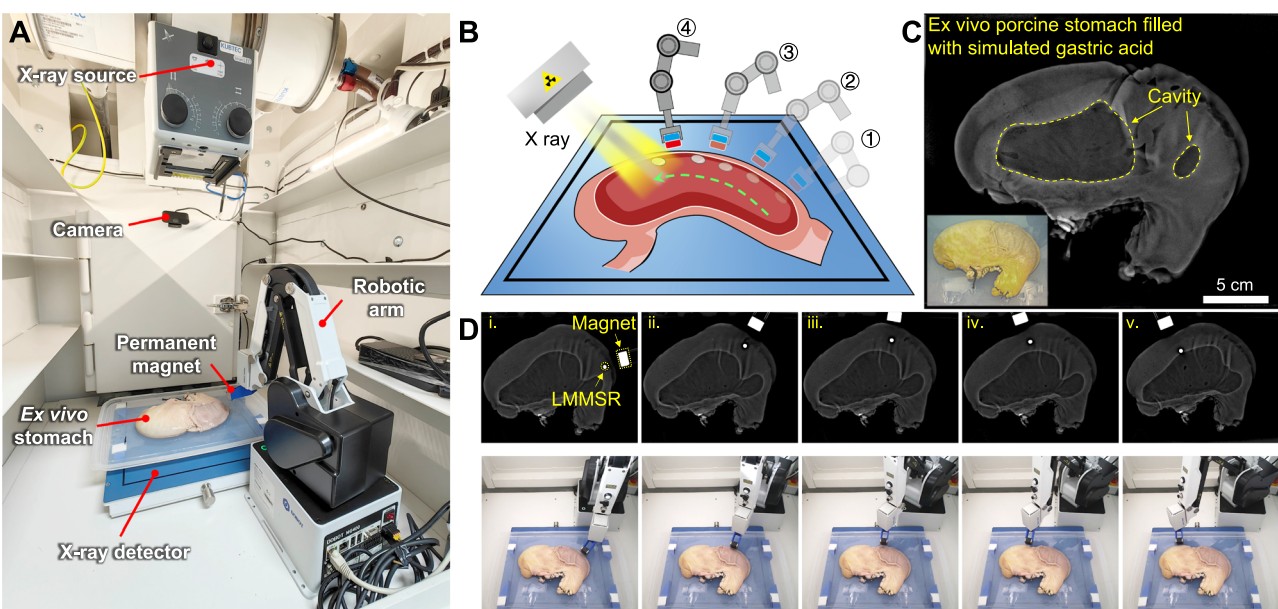

**Fig. 7 | Controllable navigation of LMMSR in an ex vivo porcine stomach under the monitoring of X-ray imaging system. A** The setup for LMMSR control experiment, which consists of an ex vivo porcine stomach, magnetic control system (a permanent magnet integrated on robotic arm), and imaging system (X-ray source and detector, and optical camera). **B** Illustration of magnetic navigation of LMMSR with a robotic magnet. **C** X-ray image of an ex vivo pristine stomach filled with simulated gastric acid, with the inset showing the optical image. **D** Sequential X-ray images and corresponding video images for remote magnetic manipulation of LMMSR.

system to observe the operation of robotic arm. Before the experiment, the pristine ex vivo stomach was first imaged, as shown in Fig. 7C, whose internal ravines could be clearly observed, while the cavity represented the region incompletely filled with acid solution. Then with the assistance of robotic arm, the LMMSR was in situ steered by external magnetic field, and the motion trajectory could be well controlled (Fig. 7D). It could be found that the signals of LMMSR and magnet were much brighter than that of tissue due to their high density-endowed radiopaque property. In contrast, conventional ferrofluids based on water, alkane and mineral oil could not be distinguished from the tissue under X-ray (Fig. S36). Therefore, we demonstrated that the biocompatible LMMSR could be employed for targeted delivery in the acidic environment of ex vivo porcine stomach under the guidance of endoscope and X-ray imaging system, which promised major benefits for future medical imaging-guided minimally invasive surgery.

## Discussion

Here, we propose a composite preparation strategy to eliminate the substantial mismatch in surface energy and improve the wettability between LM and metallic oxide, leading to the facile and rapid combination between EGaIn and $Fe_3O_4$ nanoparticles via multiple amalgamation methods. After regulating the wettability of $Fe_3O_4$ nanoparticle by introducing an intermediate silver layer, the alloying reaction between Ag and In components induced the extension and contraction of EGaIn droplet on the FPA membranes. Detailed characterization results indicated that the generated $Ag_xIn_y$ IMCs facilitated to confine magnetic agents in the interior liquid phase of EGaIn. Thus, with the assistance of metallic bond interaction and irregular morphology of $Ag_xIn_y$ IMC, iron oxide magnetic agents could be suspended steadily in LM under acidic environment, which still took effect when external magnetic force was applied. Through programming external magnetic field, the prepared LM composite exhibited a variety of controllable transformation and locomotion behaviors (passive/active deformation, splitting, fusion, migration, etc.), which was further demonstrated to be capable of performing targeted cargo transportation in vitro. Finally, we verified the feasibility of applying the developed biocompatible LMMSR in an ex vivo porcine stomach under the monitoring of endoscopy and X-ray imaging, taking a step toward the realization of clinical applications of LM-based miniature soft robot.

In the future, the proposed strategy will facilitate to improve the wetting conditions between LM and a range of inorganic non-metallic materials, and stimulate the development of versatile LM composites. It is worth noting that the prepared EGaIn-FPA microparticles developed for biocompatibility evaluation also possess the features of LM (e.g., flexibility, conductivity, photothermal property, and antibacterial property), which can be potentially applied in various areas, such as flexible electronics, active micro-scale welding and nanomedicine. Furthermore, magnetic soft robots based on the emerging functional LMs provide metal-based platform with autonomous movement ability for clinical translation of active metal stent removal, biofilm eradication and so forth.

## Methods
### Materials and characterization
In the process of fabrication for various FPA magnetic particles and magnetic liquid metal composites, EGaIn ($Ga_{75.5}In_{24.5}$, 99.999%, Dongguan Dingguan Metal Technology Co., Ltd), $FeCl_3·6H_2O$ (99%, Tianjin Baishi Chemical Co., Ltd), $FeCl_3$ (Macklin), $GaCl_3$ (Macklin), $InCl_3$ (Macklin), $AgNO_3$ (Aladdin), ethylene glycol (98%, Aladdin), polyethylene glycol 400 (Aladdin), sodium acetate (99%, Macklin), Tris-HCl (Roche Diagnostics), dopamine hydrochloride (98%, Aladdin), silver nitrate standard solution (0.1 N, Aladdin), D-glucose (AR, Macklin), ammonia water (25–28%, Aladdin), NaCl

(99.5%, Macklin), indium powder (99.9%,Topp metal), silver powder (99.9%, Leber), water-based ferrofluid (from Ink King), mineral-based ferrofluid (from Ink King), and alkane-based ferrofluid (from Ink King) were commercially purchased and used as received. In the process of preparation for EGaIn-FPA microparticles, polyvinylpyrrolidone (PVP, K29-K32, Macklin) was dissolved in the ethanol to stabilize EGaIn-FPA microparticles. In the cytotoxicity testing, Dulbecco's modified eagle medium (DMEM), trypsin, antibiotic, fetal bovine serum (FBS), and phosphate-buffered saline (pH 7.2, PBS) were commercially purchased from Thermo Fisher Scientific; Cell Counting Kit-8 (Zeta-life), Calcein-AM/PI double dye kit (Beyotime Biotechnology Co., Ltd) and glutaraldehyde (50%, Aladdin) were commercially purchased and used as received. The N52 permanent magnets were purchased from Shenzhen Lala magnetic material development Co., Ltd. The porcine stomach was purchased from a local fresh food market.

FT-IR spectra and XRD data were obtained by X-ray diffractometer (D/Max-2500/PC) and Fourier-transformed infrared spectrometer (Nicolet380, Thermo), respectively. The size distribution was measured by dynamic light scattering with Zetasizer Nano ZEP. SEM images and fluorescence images were captured by Phenom Microscopy (ProX) and Leica inverted optical microscope (DMi8), respectively. TEM images and corresponding EDX spectroscopy were obtained by Talos F200x. Rheological properties and magnetic properties were tested by rheometer (Anton Paar, MCR72) and vibrating sample magnetometer (LakeShore7404), respectively. The contact angles were measured by contact angle meter (JC2000D, Shanghai Zhongchen Digital Technology Equipment Co., Ltd). The optical videos and images showing the manipulation in an ex vivo porcine stomach were captured by endoscope (Shenzhen Changsite Technology Co., Ltd). The micro-CT analysis was characterized by three-dimensional X-ray microscopy (Zeiss Versa 515). The amounts of heavy metal elements were measured by inductively coupled plasma optical emission spectra (ICP-OES, PerkinElmer Avio 200). The cell viabilities were quantitively analyzed by microplate reader (Infinite F50, Tecan). X-ray images were obtained by XCELL 320 irradiator system (KUBTEC SCIENTIFIC). The no-current magnetic field model in commercial finite element software COMSOL Multiphysics was used to simulate the magnetic field of N52 NdFeB permanent magnet with different shapes.

### Preparation of $Fe_3O_4$@PDA@Ag nanoparticles
According to the previous research[49], $FeCl_3·6H_2O$ (9.72 g, 36 mmol) was dissolved in ethylene glycol (150 mL) to form a yellow solution, followed by the addition of sodium acetate (10.8 g, 132 mmol) and polyethylene glycol 400 (3 g). Then the mixture in a 250 mL beaker was stirred for 2 h by a magnetic stirrer (1000 r/min). Subsequently, the yellow viscous mixture was sealed in a Teflon-lined stainless-steel autoclave, which was heated to 200°C and maintained for 6 h. Black products were pooled into another beaker until the autoclave was cooled to room temperature. Finally, the products were washed several times with deionized water and ethanol and dried at 60°C overnight.

Prepared $Fe_3O_4$ nanoparticles (1 g) were dispersed in the Tris-HCl buffer (500 mL, 1 mM, pH 9) by ultrasound to form a black suspension. Then, dopamine (500 mg, Aladdin) was added to the suspension, followed by stirring and ultrasound for 4 h at room temperature. Finally, these products were washed several times with deionized water and ethanol and dried at 60 °C overnight.

According to the silver mirror reaction, by mixing moderate ammonia water and $AgNO_3$ standard solutions, silver ammonia solutions (100 mL) with different $Ag^+$ reactive concentrations were prepared firstly, including 5 mM, 10 mM, 15 mM, 20 mM, 40 mM, 60 mM, 80 mM, and 100 mM. Then, $Fe_3O_4$@PDA nanoparticles (100 mg) and excessive D-glucose (1 g) were dissolved in the above various silver ammonia solutions by mechanical stirring for 30 min (500 r/min).

Finally, these FPA particles were washed with deionized water and ethanol and dried at 60 °C overnight for utilization later.

## Preparation of EGaIn-FPA magnetic liquid metal composites and EGaIn-FPA microparticles

EGaIn (500 mg) was ground with various FPA (20 mM, 100 mM) in a natural agate mortar for several minutes, where the FPA mass ratios $m_{FPA}/m_{EGaIn}$ are controlled to be 1%, 2%, 3%, 4%, 5%, 10%, 15%, and 20%, preparing the EGaIn-FPA magnetic liquid metal composites. Moreover, the electrochemical method and acid-facilitated amalgamation could be applied to preparing this composite due to the wettability between Ag and EGaIn. The steps for the electrochemical method were as follows: An EGaIn droplet (~500 mg) in NaCl solutions (1 M) was inserted into a negative copper wire with 5 V, resulting in the oxides of EGaIn being reduced so that the droplet could wet and internalized FPA (100 mM, ~10 mg). Similarly, the process of acid-facilitated amalgamation could be briefly described as the following steps: HCl solutions (1 M) dissolved the surface oxide film of EGaIn to expose the pure liquid metal substance, which reacted with FPA (100 mM, ~50 mg), preparing the EGaIn-FPA magnetic liquid metal composites upon vibration.

The probe ultrasound method is used widely to prepare liquid metal micro/nanoparticles. EGaIn-FPA magnetic liquid metal composites (500 mg), including EGaIn-5% FPA (100 mM) and EGaIn-5% FPA (20 mM), were added to the ethanol solution (10 mL) containing PVP (800 mg), and then the mixture was sonicated in an ice bath by probe-type sonicator (JY92, Ningbo Xinzhi Biotechnology Co., Ltd) at 240 W for 1 min. After momentary sonication, the bulk liquid metal composites were broken into numerous microparticles. To sift the liquid metal microparticles with $Fe_3O_4$ nanoparticles, we applied a magnet to attract EGaIn-FPA microparticles, obtaining magnetic liquid metal microparticle inks.

## Contact angle testing

Firstly, different kinds of magnetic particles (10 mg) in ethanol solutions (2 mL) were filtrated to enable particles to cover uniformly in filter membranes (average pore size is 0.45 μm). Then, filter membranes covered with various magnetic particles (including FPA (5 mM), FPA (10 mM), FPA (15 mM), FPA (20 mM), FPA (40 mM), FPA (60 mM), FPA (80 mM), FPA (100 mM), and $Fe_3O_4$@PDA) were prepared. Subsequently, to fix these magnetic particles in the filter membranes, we pressed them violently with a double-roll machine (MSK-2150, Shenzhen Kejing Electronics Co., Ltd). For showing the wettability of EGaIn and magnetic particles, the contact angles of EGaIn droplets (~10 μL) on these various substrates after dropping HCl solutions (~5 μL) were measured by contact angle test platform. A CCD camera (25 fps) was employed during testing processes to record contact angles.

## Stability testing for magnetic liquid metal composites

For testing the magnetic suspension stability, 500 mg EGaIn-5% FPA (100 mM) and 500 mg EGaIn-1.25% $Fe_3O_4$ prepared by mechanical grinding were immersed in HCl solutions (pH 0.5, 5 mL) for 24 h, where magnets (20 mm × 10 mm × 3 mm, $B_{ext}$ = 200 mT) were set to attract $Fe_3O_4$ in two composites. It should be noted that the initial amount of $Fe_3O_4$ in both composites was generally equivalent based on the VSM data. Finally, the leakage of $Fe_3O_4$ was quantified by the amount of Fe element in the supernatant measured by ICP-OES after removing these treated magnetic liquid metal composites. The experiments were repeated three times to reduce errors.

For testing the stability of EGaIn-5% FPA (20 mM) in the simulated gastric acid, 500 mg EGaIn-5% FPA (20 mM) were immersed in HCl solutions (pH 1, 1 mL) for 15 min, 30 min, 60 min, and 180 min, respectively. Then, the amount of released heavy metal elements (Ag, Ga, In, and Fe) in the supernatant was measured by ICP-OES after

removing treated EGaIn-5% FPA (20 mM). The experiments were repeated two times to reduce errors.

## Deformation performance testing for LMMSR

Firstly, the effect of magnetic agent fraction on active deformation was studied. Three experimental groups were set up, and the mass ratio of FPA (100 mM) were controlled to be 1%, 3% and 5%, respectively. Other conditions were the same. We poured an appropriate amount of HCl solution (1 M) and LMMSR with different FPA mass ratio (0.5 g) into a square transparent glass cuvette, and placed a NdFeB permanent magnet (N52, 24 mm × 24 mm × 12 mm, $B_{ext}$ = 500 mT), recording the active deformation under the action of an external magnetic field.

Then, the effect of the HCl concentration on active deformation was investigated. The HCl concentrations were controlled to 0.05 M, 0.1 M, 0.2 M, 0.5 M, 0.8 M and 1 M, respectively, and other conditions were the same. LMMSR (EGaIn-5% FPA (100 mM), 0.5 g) and hydrochloric acid solutions with different concentrations designed in advance were poured into a square glass cuvette, recording the aspect ratio of LMMSR with and without an external magnetic field.

Next, the influence of the external magnetic field strength on active deformation was studied. We poured the LMMSR (EGaIn-5% FPA (100 mM), 0.5 g) and HCl solution (1 M) into the square glass cuvette. Then we controlled the magnetic field strength by continuously adding glass slides with a thickness of 1 mm between magnet and LMMSR, observing the active deformation under different external magnetic fields.

Subsequently, we studied the passive deformation ability of LMMSR to pass through narrow channels under the actuation of magnetic field. Narrow channels with various widths were prepared (including 9 mm, 8.5 mm, 8 mm, 7.5 mm, 7 mm, 6.5 mm, 6 mm, 5.5 mm, 5 mm, 4.5 mm, 4 mm, 3.5 mm, 3 mm, 2.5 mm, 2 mm, 1.5 mm, and 1 mm) using 3D printers (e.g., NanoArch S130, BMF Precision, China), and the magnet was used to control LMMSR to pass through these narrow channels with HCl (1 M) from wide to narrow successively. Additionally, we changed FPA (100 mM) mass ratio (1%, 2%, 3%, 4% and 5%) and strength of external magnetic field to test their influences for passive deformation ability.

Finally, to test the duration of LMMSR, we first used a magnet to drive the LMMSR built by EGaIn-5% FPA (100 mM) back and forth along a distance of 60 mm in HCl solution (1 M) with a speed of ~10 cm/s. After the actuation cycle number reached 10, 50, 100, 150, and 200, respectively, the LMMSR was transferred to narrow channels with various widths to test passive deformation behaviors. The same experiments were conducted for the LMMSR after 1 day, 3 days, and 7 days.

## Actuation performance stability testing for LMMSR

To test the duration of LMMSR for actuation performance, we used a permanent magnet to repeatedly drive EGaIn-5% FPA (100 mM) liquid metal composite-based LMMSR along a distance of 60 mm back and forth with a speed of 10 cm/s in hydrochloric acid solution (1 M). After a certain number of cycles (10, 50, 100, 150, and 200), LMMSR was transferred to a straight tube, and a permanent magnet (N52, 24 mm × 24 mm × 12 mm, $B_{ext}$ = 500 mT) below tube was controlled to gradually approach LMMSR. When the distance between LMMSR and magnet reached a critical value, the magnetic attractive force would become strong enough to drive LMMSR towards magnet. Such value was defined as maximum response distance of LMMSR, and the average driving velocity of LMMSR towards magnet was also recorded.

## In vitro cytotoxicity of magnetic liquid metal composite and various ferrofluids

3T3 cells and HEK293 cells with a density of 3000 cells per well were seeded in a 96-well plate, followed by 12 h incubation in 100 μL DMEM

with 10% FBS and 1% antibiotic. Then, various 100 µL fresh mediums containing different concentrations of EGaIn-5% FPA (20 mM) microparticles, were added to the wells incubating 3T3 and HEK293 cells after the initial medium was discarded. Subsequently, these samples were co-cultured with cells for 24 h. Next, after the second medium was discarded under magnetic field to avoid the magnetic particles from being removed, 100 µL fresh medium containing 10 µL CCK-8 was added to each well, followed by another 2 h incubation. To remove the extra light absorption value of EGaIn-5% FPA (20 mM) microparticles, we added the medium with microparticles to another 96-well plate without cells, where the arrangement of the concentration gradient was the same as the 96-well plate with cells. The absorbance was detected at 450 nm with a microplate reader. All of the tests were repeated eight times to reduce errors.

HEK293 cells were seeded in a 96-well plate at a density of 3000 cells per well and incubated for 12 h in 100 µL DMEM with 10% FBS and 1% antibiotic. After discarding the initial medium, different fresh mediums containing different concentrations (12, 25, 50, 100, 200 µg/mL) of mineral oil-based and alkane-based ferrofluids were added to the wells. The samples were then co-cultured with cells for 24 h. Next, the wells were washed three times with 100 µL PBS solution. A fresh medium containing 10 µL CCK-8 was added to each well, followed by another 2-h incubation. To remove the extra light absorption value of magnetic particles, the medium containing CCK-8 was transferred to a new 96-well plate after chromogenesis, and a magnet was used to avoid the removal of magnetic particles. Finally, the absorbance was detected at 450 nm using a microplate reader. All tests were repeated eight times to reduce errors.

### Live/dead cell staining assay

3T3 and HEK293 cells were seeded onto a 96-well plate at a density of 5000 cells per well and cultured with medium for 12 h. After cell attachment, we replaced 100 µL medium with 20 µg EGaIn-5% FPA (20 mM) microparticles to the initial medium, followed by another 24 h incubation. Then, the cells were washed three times with PBS and stained with Calcein-AM (5 µL) and PI (5 µL) solution in the PBS buffer (5 mL) for 30 min at room temperature after the medium was removed. Finally, both cells were washed with PBS several times, and staining results were observed by inverted fluorescence microscope at 494 nm and 535 nm.

### In vitro cytotoxicity of leaked metal ions

We measured cytotoxicity of the metal element concentration when immersing LMMSRs in simulated gastric acid for 1 h. Anhydrous $FeCl_3$, $GaCl_3$, $InCl_3$, and $AgNO_3$ powders were used to prepare nutrient media with the leaked metal ions (Ga = 17.40 mg/L, In = 1.90 mg/L, Fe = 33.52 mg/L, Ag = 3.94 mg/L). HEK293 cells (3000 per well) were co-cultured with the prepared nutrient media in 96-well plates for different times (2, 4, 6, and 12 h). After incubation, the cells were washed three times with PBS, and 100 µL DMEM medium containing 10 µL CCK-8 was added. After 2-3 h, a microplate reader was used to test the chromogenic solution in the 96-well plate under light with a wavelength of 450 nm. All tests were repeated three times to reduce errors.

### Gradient dehydration for cells

HEK293 cells (50000 cells per well) were seeded onto a 6-well plate where glass sheets were placed, followed by 12 h incubation in 1 mL medium. Then, 200 µg EGaIn-5% FPA (20 mM) microparticles were added to wells, co-culturing with HEK293 cells for 4 h. After removing the medium and washing cells several times with PBS, glutaraldehyde solutions (2.5%, 1 mL) were added to the well to fix the cell structure for another 2 h. Next, various ethanol aqueous (30%, 50%, 75%, 80%, 95%, and 100% ethanol) were added to these wells by turn, in which, the duration of every dehydration was about 15 min.

### Ex vivo magnetic manipulation for LMMSR

A LMMSR (~400 µL) based on EGaIn-5% FPA (20 mM) was injected into a porcine stomach containing 20 mL simulated gastric acid. Meanwhile, a capsule loaded with dye was arranged to stick in the gastric mucosa. A permanent magnet (N52, NdFeB, 24 mm by 24 mm by 12 mm) was utilized to control the LMMSR. The distance between the magnet and the LMMSR is about 5 mm. Moreover, to achieve accurate manipulation and real-time observation, we applied an endoscope to observe the behaviors of LMMSR upon targeted transportation.

During X-ray imaging, we applied a permanent magnet fixed on a robotic arm (Dobot MG400) to remotely control a LMMSR inside a sealed ex vivo stomach filled with stimulated gastric acid. The locomotion trajectory for LMMSR is preset from the pylorus to the greater curvature.

### Reporting summary

Further information on research design is available in the Nature Portfolio Reporting Summary linked to this article.

## Data availability

The authors declare that the data supporting the findings of this study are available within the paper and its supplementary information files. Source data are provided with this paper.

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

## Acknowledgements

The work is financially supported by National Natural Science Foundation of China with a No. of 92163109 (X.M.), Shenzhen Science and Technology Program with Nos. of JCYJ20200109113408066, and KQTD20170809110344233 (X.M.), and the Fundamental Research Funds for the Central Universities with a Grant No. of HIT.OCEF.2021032 (X.M.). D.J. thanks the support from National Natural Science Foundation of China with a No. of 52202348 (D.J.), Guangdong Basic and Applied Basic Research Foundation with a No. of 2023A1515011491 (D.J.), and Shenzhen Science and Technology Program with a No. of GXWD20220818224716001 (D.J.). The authors thank Dr. Fengtong Ji for his help in magnetic field simulation. The authors also thank Liying Wang and Dongfang Zhao for their assistance in manuscript revision.

## Author contributions

Y.S., D.J., and X.M. conceived the project, and D.J. and X.M. supervised the studies. Y.S. prepared the magnetic liquid metal composites and performed experiments for behaviors of liquid metal magnetic soft robots. M.F. helped to perform the cellular experiments and contact angle measurements. Sa.L. and Q.C. helped to perform the ex vivo porcine stomach experiments. Y.S. analyzed the data. Z.X., B.W., G.L., W.C., and Sh.L. provided suggestions for the work. Y.S., D.J., and X.M. co-wrote the manuscript. All authors have given approval to the final version of the manuscript.

## Competing interests

The authors declare no competing interests.
