## [Peer Review File · Nature Communications]

Reviewers' comments:

Reviewer #1 (Remarks to the Author):

Yifeng Shen et al. have reported a method of magnetic gallium-based room temperature liquid metal composite capable of incorporating functionalized magnetic materials for magnetic actuation of the droplets. Magnetic actuation is a trending topic and has been done previously. The anchoring method through an interlayer PDA and incorporation of these particles into the liquid metals are new, and the application is also intriguing. The magnetic actuation has been comprehensively engineered and characterized in this paper. The paper is high quality and interesting. The content, however, can be enhanced and clarified further and appropriate articles should be cited. My comments are as below and the paper can be published after minor revisions:

1. The anchoring mechanism is fundamentally novel. The authors should also clarify and include relevant surface layering of EGaIn such as the outermost layer of this complex is covered by Indium metal after the removal of the gallium oxides. Surface Indium enrichment effectively enables intermetallic formation with Ag during the contact. Otherwise, there are also reports including intermetallic formation between Ag and Ga if Ag is first alloyed into the InGa complex. Cite Phys. Rev. B 55, 15874–15884 (1997) and Surface Science Volume 124, Issues 2–3, 2 January 1983, Pages 407-422. Is this a reaction competition between related to energy or an interfacial phenomenon citing Science 2017, 358, 6361 pp. 332-335. Authors mentioned reaction preference. Otherwise, In-Ag and Ga-Ag reaction energies should be cited.

2. Authors are encouraged to perform a control experiment with pure Ga in the absence of In. and also, in the second experiment, Add Ag to EGaIn first before anchoring, all followed by XRD (supporting comment 1). The referenced 40 did not include intermetallic formation preference use other supporting references, include discussion in comment 1 or perform 2 control experiments which can be characterized using XRD.

3. Cite relevant magnetic actuation of liquid metals: Lab Chip, 2017, 17, 128-133, Nano Lett. 2022, 22, 7, 2923–2933, CS Appl. Mater. Interfaces, Nat Commun 10, 1300 (2019), Journal of Materials Science & Technology Volume 92, 30 November 2021, Pages 60-68, Nano Letters, vol. 17, no. 12, pp. 7831-7838. Cite other relevant shape-shifting methods and self-propulsion, including the deformation mechanism supplied Nat Commun 7, 12402 (2016)

4. The mention of jump and bouncing effects should be clarified both in text and videos. There is no visible separation of droplet and surface during the jump. Authors should use other terminology if there is no separation.

5. Authors have measured stability using ICP when the droplet is simply placed inside the partly acceptable solvent. However, does the actuation actually make a difference, for example, by separating solid particles? Some of the videos show particle separation. This should be clarified and characterized after actuation cycles and also looking at separated solids, not only dissolved solids. This is a competing argument between surface functionalization and simply alloying magnetic

particles inside the droplet, in which the surface functionalization is usually criticized as less stable. The authors should clarify this by characterizing the separation of any nanoparticles after actuation.

Reviewer #2 (Remarks to the Author):

This paper reports magnetic liquid metal composites made by a modified magnetic agent (iron oxide particles) that can maintain their softness, magnetism, and stability, contrary to conventional magnetic liquid metal composites. Such magnetically responsive liquid metals are of interest for a variety of proposed applications, including the use of magnetic fields to move / manipulate the composite material (liquid metals by themselves are diamagnetic). Normally, iron oxide will not mix into liquid metal. The authors address this challenge by coating iron oxide with Ag, which can promote reactive wetting with the liquid metal. I am not surprised by the work as reactive wetting is well-known, but to the best of my knowledge, it has not been used in this exact way.

1) Page 2, Line 80 - 83:

The authors mention that the most important issue is to address interfacial non-wettability caused by the substantial mismatch in surface energy, but this statement seems quite vague. Does the interface mean an interface between non-metallic materials and liquid metal, or what? Please rephrase the statement to make it easier to understand.

2) Page 3, Line 95:

The mention of a reactive wetting mechanism can be confusing. The sentence can be interpreted as saying that oxide (Fe_3O_4) and LM react, so please be more explicit.

3) Page 7, the first passage in the Anchoring effect section:

a) In the mechanical grinding case, did you conduct the grinding process of LM and modified Fe_3O_4 powders in the air or glovebox?

b) If the process was conducted in the air, was there a difference in the viscosity of samples between the three magnetic liquid metal preparation methods, mechanical, electrochemical, and acid-facilitated? Please provide data if there is a difference in the fabricated magnetic LM composite properties between the three preparation methods.

4) Page 7, the second passage in the Anchoring effect section:

The authors mention the advantage of the modified magnetic particles capped by Ag in terms of encapsulation of Fe₃O₄. However, in terms of durability, the alloying phenomenon between Ag and Indium could cause the excess alloy to deteriorate the property of magnetic LM composites, such as viscosity, or phase state (like the composite becomes turns into a solid state from a liquid state. Is there data about the time-dependence durability of the magnetic LM composite?

5) Page 8 and Figure 4, the first passage in the deformation section:

The authors mention that the fabricated sample with Fe₃O₄@PDA@Ag still has good viscosity even though the sample is a composite. However, will the fabricated sample via reactive wetting retain its viscosity over time? (an extension of the question about the excess alloying of Ag-In)

Reviewer #3 (Remarks to the Author):

A significant portion of the research article and the primary novelty focuses on overcoming the intrinsic interfacial non-wettability between the Fe₃O₄ particles and EGaIn using an intermediate silver layer via polydopamine to facilitate reactive wetting and embed/anchor the Fe₃O₄ nanoparticles inside the EGaIn. However, this strategy has already been introduced by other researchers such as Ref 35. While this work is referenced by the authors, these prior contributions are largely ignored by the authors and the scientific advancement over this work is unclear. The authors contribution is largely incremental and not suitable for publication in Nature Communications.

Response to Reviewer #1

Yifeng Shen et al. have reported a method of magnetic gallium-based room temperature liquid metal composite capable of incorporating functionalized magnetic materials for magnetic actuation of the droplets. Magnetic actuation is a trending topic and has been done previously. The anchoring method through an interlayer PDA and incorporation of these particles into the liquid metals are new, and the application is also intriguing. The magnetic actuation has been comprehensively engineered and characterized in this paper. The paper is high quality and interesting. The content, however, can be enhanced and clarified further and appropriate articles should be cited. My comments are as below and the paper can be published after minor revisions

Response: Thanks a lot for your positive comments. We have answered every comment and made corresponding changes in the manuscript. Please check the following point-by-point response for details.

1. The anchoring mechanism is fundamentally novel. The authors should also clarify and include relevant surface layering of EGaIn such as the outermost layer of this complex is covered by Indium metal after the removal of the gallium oxides. Surface Indium enrichment effectively enables intermetallic formation with Ag during the contact. Otherwise, there are also reports including intermetallic formation between Ag and Ga if Ag is first alloyed into the InGa complex. Cite Phys. Rev. B 55, 15874–15884 (1997) and Surface Science Volume 124, Issues 2–3, 2 January 1983, Pages 407-422. Is this a reaction competition between related to energy or an interfacial phenomenon citing Science 2017, 358, 6361 pp. 332-335. Authors mentioned reaction preference. Otherwise, In-Ag and Ga-Ag reaction energies should be cited.

Response: Thank you very much for your valuable suggestions. We totally agree with the reviewer that the surface indium enrichment effect might facilitate the preferential formation of Ag_xIn_y intermetallic compounds (IMCs) when magnetic agents ($\text{Fe}_3\text{O}_4@\text{PDA}@\text{Ag}$, FPA) came into contact with EGaIn liquid metal. However, indium atoms in the extremely thin enrichment layer could hardly exhaust the silver atoms on the surface of FPA immediately upon mechanical grinding (a relatively rigorous mixing process). As a result, the inner bulk EGaIn component should inevitably expose to the FPA particles, and at this time, direct contact between Ga, In

and Ag occurred. In our work, Ag_xIn_y rather than Ag_xGa_y IMCs were detected in the
liquid metal composite, and we attributed such phenomenon to the reaction preference
between In and Ag.

To support our hypothesis, we conducted several control experiments according to
your suggestion. In the first one, we mixed FPA (100 mM) particles with pure Ga
liquid metal via mechanical grinding, and the corresponding XRD characterization
was presented in **Fig. R1.1**. The presence of prominent Ag_2Ga peaks indicated that Ag
shell of FPA could indeed react with gallium to produce Ag_xGa_y IMCs in the absence
of indium. Then, in the second control experiment, we ground Ag particles with
EGaIn liquid metal. The XRD results in **Fig. R1.2A** indicated that with the increase of
Ag particle amount, the formed IMC phase in liquid metal composite gradually
changed from AgIn_2 to Ag_9In_4 . While Ag_xGa_y IMC did not occur even when the mass
ratio between Ag and EGaIn was as high as 50% (i.e., mixing 10 g EGaIn with 5 g Ag
particle). In the third control experiment, 0.5 g Ag particle was first mixed with 10 g
pure Ga liquid metal, followed by the addition of 1 g In powder. In **Fig. R1.2B**,
Ag_2Ga phase was initially identified in Ga-Ag mixture by XRD characteristic peaks.
However, Ag_2Ga peaks disappeared after In powder was added, while new peaks
corresponding to Ag_9In_4 phase appeared instead, indicating Ag atoms were more
inclined to bind with In atoms compared to Ga atoms. Therefore, in the Ga-In-Ag
ternary system, reaction preference between In and Ag existed, which could be the
reason why only Ag_xIn_y IMCs were detected when we mixed EGaIn liquid metal with
FPA particles in our work. Such comparison experiments have been added to the
revised Supporting Information as Fig. S16-S18.

**Fig. R1.1.** XRD pattern of Ga-10% FPA(100 mM) composites, where perpendicular
pink lines and the insert represent the standard diffraction peaks and the crystal
structure of Ag_2Ga , respectively.

**Fig. R1.2.** (A) XRD patterns of EGaIn liquid metal composites with various mass
 fraction of Ag. Standard PDF cards of Ag_3In_4 , Ag_3In , AgIn_2 and Ag_2Ga are provided
 for comparison. (B) XRD patterns of Ga-Ag composite before and after In addition.

Besides, from the thermodynamic viewpoint of In-Ag and Ga-Ag systems [*J. Chem.*
 *Thermodyn.* 2011 43, 392-398] [*Thermochim. Acta.* 2005, 433, 66-71], the logarithm
 of activity coefficients γ at infinite dilution under 1200 K yielded $\ln \gamma_{\text{Ga}}^0 = -2.24$,
 and $\ln \gamma_{\text{In}}^0 = -3.16$, respectively. In turn, comparison of the enthalpy of mixing in
 these two systems showed that $\Delta_{\text{mix}}H_m$ in In-Ag alloy was generally lower than that in
 the Ga-Ag system, i.e., interactions between Ag and In atoms were stronger than those
 between Ag and Ga atoms. Therefore, in the Ga-In-Ag ternary system, Ag_xIn_y phase
 was preferentially formed. Such phenomenon was consistently observed when silver
 was mixed with EGaIn liquid metal at room temperature [*Nat. Electron.* 2023, 6, 206-
 215] [*Adv. Mater.* 2018, 30, 1801852] [*ACS Appl. Mater. Interfaces* 2021, 13, 14552-
 14561]. Furthermore, similar reaction competition had been reported in the Al-Mn-Ga
 ternary alloy [*Science* 2022, 378, 1118-1124]. In their work, through introducing Al
 into Mn-Ga system, the original MnGa phase disappeared, and was replaced by
 $\text{Al}_{11}\text{Mn}_4$ phase with relatively lower Gibbs energy of formation.

In summary, despite the existence of indium enrichment layer, we think the reaction
 competition between In and Ga with Ag should be the main reason why only Ag_xIn_y
 IMCs were detected in our EGaIn-FPA composite. We have added such description to
 the revised manuscript and supporting information, and cited relevant works as Ref
 [61-63].

2. Authors are encouraged to perform a control experiment with pure Ga in the
 absence of In. and also, in the second experiment, Add Ag to EGaIn first before
 anchoring, all followed by XRD (supporting comment 1). The referenced 40 did not

include intermetallic formation preference use other supporting references, include
discussion in comment 1 or perform 2 control experiments which can be characterized
using XRD.

**Response:** Thanks for your comment. According to your suggestion, we conducted
three control experiments, whose details have been introduced in the response to
comment 1. When silver modified magnetic agents FPA were mixed with pure Ga in
the absence of In, Ag₂Ga XRD characteristic peaks were detected, indicating that Ag
shell of FPA could indeed react with gallium to produce Ag_xGa_y IMCs. Then, through
adding Ag particles to EGaIn, the phase of formed IMCs gradually changed from
AgIn₂ to Ag₉In₄ with the increase of Ag mass fraction. While Ag_xGa_y IMCs did not
occur even when Ag mass fraction reached 50%, which was much higher than that in
our EGaIn-FPA composite. Finally, we mixed Ag particle with pure Ga, followed by
the addition of In powder. It was found that the initially formed Ag₂Ga phase was
completely replaced by Ag₉In₄ phase after In powder was added to Ga-Ag system. All
the experiments indicate the existence of reaction competition between Ga and In with
Ag, in which, Ag preferentially bonds with In to form Ag_xIn_y IMCs. We have removed
Ref [40] of the original manuscript, and cited relevant works as Ref [65, 66] in the
revised manuscript.

3.Cite relevant magnetic actuation of liquid metals: *Lab Chip*, 2017,17, 128-133,
*Nano Lett.* 2022, 22, 7, 2923–2933, *ACS Appl. Mater. Interfaces*, *Nat Commun* 10,
1300 (2019), *Journal of Materials Science & Technology* Volume 92, 30 November
2021, Pages 60-68, *Nano Letters*, vol. 17, no. 12, pp. 7831-7838. Cite other relevant
shape-shifting methods and self-propulsion, including the deformation mechanism
supplied *Nat Commun* 7, 12402 (2016)

**Response:** Thanks for your comment. According to your suggestion, we have cited
these relevant papers in the revised manuscript.

*Lab Chip* 2017,17, 128-133 as Ref [39].

*Nano Lett.* 2022, 22, 2923-2933 as Ref [40].

*Nat. Commun.* 2019 10, 1300 as Ref [31].

*J. Mater. Sci. Technol.* 2021, 92, 60-68 as Ref [32].

*Nano Lett.* 2017, 17, 7831-7838 as Ref [33].

*Nat. Commun.* 2016, 7, 12402 as Ref [30].

4. The mention of jump and bouncing effects should be clarified both in text and
videos. There is no visible separation of droplet and surface during the jump. Authors
should use other terminology if there is no separation.

**Response:** Thanks a lot for your comment. In the original manuscript, the aim of
using “jump” and “bounce” terms was to illustrate the wettability transition between
the liquid metal droplet and FPA membrane. But actually, no visible separation
between droplet and membrane could be obtained as the reviewer stated. We are sorry
for causing confusion. In the revised manuscript, we remove “jump” and “bounce”
terms, and then use “extension” and “contraction” to describe the dynamic process of
liquid metal on FPA membrane. Similar process under an external electronic field has
been reported to use these terminologies [*Adv. Mater.* 2021, 33, 2103062], and we
hope such revision can help to avoid potential misunderstanding.

5. Authors have measured stability using ICP when the droplet is simply placed inside
the partly acceptable solvent. However, does the actuation actually make a difference,
for example, by separating solid particles? Some of the videos show particle
separation. This should be clarified and characterized after actuation cycles and also
looking at separated solids, not only dissolved solids. This is a competing argument
between surface functionalization and simply alloying magnetic particles inside the
droplet, in which the surface functionalization is usually criticized as less stable. The
authors should clarify this by characterizing the separation of any nanoparticles after
actuation.

**Response:** Thank you very much for your valuable comments. To evaluate the effect
of actuation on the stability of liquid metal magnetic soft robot (LMMSR), we used a
permanent magnet to repeatedly drive EGaIn-5% FPA (100 mM) liquid metal
composite-based LMMSR along a distance of 60 mm back and forth with a speed of
10 cm/s in hydrochloric acid solution (1 M). After a certain number of cycles (10, 50,
100, 150, and 200), the leaked magnetic solids were collected and characterized.

Firstly, the separated magnetic solids were dissolved in aqua regia, whose content was
further measured with ICP-OES. Then the mass proportion of leaked magnetic agents
was calculated and displayed in **Fig. R1.3**, which was found to be extremely limited

with the increase of cycles. Only 2.09% magnetic nanoparticles leaked out after 200
actuation cycles. Furthermore, the morphology of separated solid nanoparticles was
shown in **Fig. R1.4A**. The outermost Ag shell of FPA disappeared due to the corrosion
of EGaln, and the coarse surface of bare Fe₃O₄ nanoparticles were exposed.
Additionally, as illustrated in the SEM and corresponding EDX mapping results (**Fig.**
**R1.4B**), Ag_xIn_y IMC was found to be formed on the surface of magnetic particles,
indicating the reaction between Ag and In to anchor magnetic nanoparticles into liquid
metal. Such comparison experiments have been added to the revised Supporting
Information as Fig. S24 and Fig. S25.

**Fig. R1.3.** Mass proportion of leaked magnetic agent as a function of the number of
magnetic actuation cycles.

**Fig. R1.4.** Characterization of the separated magnetic solids after actuation cycles. (A)
SEM image of the leaked magnetic agent. (B) SEM image and EDX mappings
showing the appearance of leaked magnetic agents with a surface coating of Ag_xIn_y
IMC.

Besides, when the number of actuation cycles reached 10, 50, 100, 150, and 200,
 respectively, LMMSR was transferred to evaluate its stability in magnetic actuation
 performance. As shown in the left inset of **Fig. R1.5A**, the transferred LMMSR was
 placed in a straight tube, and a permanent magnet (surface magnetic field ~ 500 mT)
 below tube was controlled to gradually approach LMMSR. When the distance
 between LMMSR and magnet reached a critical value, the magnetic attractive force
 would become strong enough to drive LMMSR towards magnet. Such value was
 defined as maximum response distance of LMMSR here, and the average driving
 velocity of LMMSR towards magnet was also recorded, both of which could
 demonstrate the magnetic responsiveness of LMMSR. As shown in **Fig. R1.5B** and
 **R1.5C**, although the maximum response distance and average velocity of LMMSR
 decreased after 100 actuation cycles due to the leakage of magnetic agents, the values
 still remained more than 70% of the original performances when cycle number
 reached 200.

 **Fig. R1.5.** Stability in magnetic actuation performance of LMMSR after different
 actuation cycles. (A) Illustrations showing magnetic actuation cycle and the follow-up
 magnetic actuation performance evaluation for LMMSR. (B) Maximum response
 distance of LMMSR as a function of cycle number. (C) Average velocity of LMMSR
 as a function of cycle number. (D) Passive deformation performance of LMMSR as a
 function of cycle number.

Afterwards, the LMMSR was transferred again to traverse a narrow slit via passive
 deformation under the guidance of a permanent magnet. **Fig. R1.5D** displayed the
 maximum deformation rate that LMMSR could achieve. It could be found that the

deformation rate decreased from 75% to 70% after 100 cycles, but the performance
remained at 70% as the number of cycles increased. These results indicated that the
magnetic actuation performance of LMMSR kept relatively stable and reliable after
hundreds of actuation cycles.

In summary, repeated actuation under external magnetic field would lead to a limited
leakage of magnetic particles from the EGaIn-5% FPA (100 mM) composite, which
only brought very little difference in the actuation performance of composite. Such
stability could fully satisfy the demands of magnetic miniature soft robot in our work
for targeted cargo delivery, and the corresponding results have been added to the
revised Supporting Information as Fig. S22 and Fig. S23.

Response to Reviewer #2

This paper reports magnetic liquid metal composites made by a modified magnetic
agent (iron oxide particles) that can maintain their softness, magnetism, and stability,
contrary to conventional magnetic liquid metal composites. Such magnetically
responsive liquid metals are of interest for a variety of proposed applications,
including the use of magnetic fields to move / manipulate the composite material
(liquid metals by themselves are diamagnetic). Normally, iron oxide will not mix into
liquid metal. The authors address this challenge by coating iron oxide with Ag, which
can promote reactive wetting with the liquid metal. I am not surprised by the work as
reactive wetting is well-known, but to the best of my knowledge, it has not been used
in this exact way.

**Response:** Thanks for your valuable comments. While reactive wetting mechanism is
well-established, we extended the application of this concept in our work to endow
originally non-wettable iron oxide with the capability of efficiently and effectively
mixing with EGaIn, contributing to the fabrication of liquid metal-based miniature
soft robot. We have answered every comment and made corresponding changes in the
revised manuscript. Please check the following point-by-point response for details.

1) Page 2, Line 80 - 83:

The authors mention that the most important issue is to address interfacial non-
wettability caused by the substantial mismatch in surface energy, but this statement
seems quite vague. Does the interface mean an interface between non-metallic
materials and liquid metal, or what? Please rephrase the statement to make it easier to
understand.

**Response:** Thanks for your comment. The interface indeed means the interface
between non-metallic material (Fe_3O_4 nanoparticle) and liquid metal. The huge
surface tension of liquid metal resulted in its non-wettability with a variety of
inorganic substrates. Therefore, in our work, through surface modification of Fe_3O_4
nanoparticles, silver coating was introduced to serve as the intermediate layer between
Fe_3O_4 and liquid metal, which could react with liquid metal to generate Ag_xIn_y
intermetallic compounds, and thus improve the interfacial wettability. To facilitate
reading and understanding, we have changed the description of “interface” in the
revised manuscript as follows.

(Line 81 - 82), the statement of “the most important issue is to address interfacial non-
wettability” is revised to “the most important issue is to address interfacial non-
wettability between Fe₃O₄ and LM”.

2) Page 3, Line 95:

The mention of a reactive wetting mechanism can be confusing. The sentence can be
interpreted as saying that oxide (Fe₃O₄) and LM react, so please be more explicit.

**Response:** Thanks for your suggestion. Wetting can be broadly classified into two
categories, i.e., non-reactive wetting and reactive wetting [*Adv. Colloid Interface Sci.*
2007, 133, 61-89]. A liquid spreading on a substrate with no reaction/absorption of the
liquid by substrate material is known as non-reactive or inert wetting (e.g., a water
droplet on a glass substrate). On the other hand, the wetting process influenced by
reaction between the spreading liquid and substrate material is known as reactive
wetting. In our work, actually Fe₃O₄ could not react with gallium-based liquid metal.
Liquid metal is not inclined to wet and spread on the surface of pristine Fe₃O₄
nanoparticles due to the massive mismatch of surface energy [*Small Methods* 2022, 6,
2200246]. To address this problem, silver shells were coated on Fe₃O₄ nanoparticles
to obtain new kind of magnetic agents. When EGaIn liquid metal contacted with the
modified magnetic agents, alloying reaction between In (liquid) and Ag (substrate)
occurred to produce intermetallic compound, which subsequently improved the
wettability of liquid metal on the surface of magnetic agents. Therefore, we claimed
that iron oxide-based magnetic agents were composited into liquid metal via reactive
wetting mechanism according to the definition. We have also modified the
corresponding sentences that might be confusing in the revised manuscript.

(Line 93 - 96) as an example, the sentence “magnetic LM composite that is capable of
incorporating non-wettable iron oxide magnetic nanoparticles within eutectic gallium
indium (EGaIn) via reactive wetting mechanism” is revised to “magnetic LM
composite that is capable of incorporating non-wettable iron oxide magnetic
nanoparticles within eutectic gallium indium (EGaIn) via reactive wetting between
LM and Ag shell modified Fe₃O₄ nanoparticle”.

3) Page 7, the first passage in the Anchoring effect section:

a) In the mechanical grinding case, did you conduct the grinding process of LM and
modified Fe₃O₄ powders in the air or glovebox?

**Response:** Thanks for your comment. In our work, the grinding process was mainly
conducted in the air. Furthermore, we also conducted mechanical grinding in the
glovebox filled with nitrogen gas. As shown in **Fig. R2.1**, modified Fe_3O_4 powders
with Ag shells were easy to compound with EGaln in an oxygen-free atmosphere. In
contrast, under the same circumstance, pristine Fe_3O_4 powders without Ag shell
proved challenging to wet with EGaln. Such comparison experiments have been
added to the revised Supporting Information as Fig. S13.

**Fig. R2.1.** Mechanical grinding of EGaln liquid metal with $\text{Fe}_3\text{O}_4@\text{PDA}@\text{Ag}$ (FPA
(100 mM)) (A) and Fe_3O_4 (B) powders in a glovebox filled with nitrogen gas.

b) If the process was conducted in the air, was there a difference in the viscosity of
samples between the three magnetic liquid metal preparation methods, mechanical,
electrochemical, and acid-facilitated? Please provide data if there is a difference in the
fabricated magnetic LM composite properties between the three preparation methods.

**Response:** Thanks for your comment. As reviewer suggested, we have updated
relevant viscosity data of EGaln-5% FPA (100 mM) magnetic liquid metal composites
prepared by mechanical and acid-facilitated methods, as shown in **Fig. R2.2**.
Electrochemical method was not conducted for comparison here because only a few
amounts of FPA (100 mM) powders could be composited into EGaln liquid metal
(less than 5% in weight percentage). The liquid metal composite samples exhibited

shear-thinning properties regardless of preparation methods. It was noted that the
 viscosity, storage modulus (G'), and loss modulus (G'') of EGaIn-5% FPA (100 mM)
 prepared by mechanical grinding were slightly lower than those of liquid metal
 composite prepared by acid-facilitated method. This might be due to emergence of
 pores and acid corrosion products on the surface of liquid metal composite during the
 acid-facilitated mixing process [*ACS Appl. Mater. Interfaces* 2015, 7, 23163-23171].
 In summary, there is a little difference in the rheological properties of magnetic liquid
 metal composites between different preparation methods. Such results have been
 added to the revised Supporting Information as Fig. S19, and in the revised
 manuscript, all the liquid metal composite samples were prepared by mechanical
 grinding if not specifically stated.

 **Fig. R2.2** (A) The viscosity of EGaIn-5% FPA (100 mM) prepared by mechanical
 grinding and acid-facilitated methods as a function of various shear rates. (B) The
 dynamic storage and loss modulus of EGaIn-5% FPA (100 mM) prepared by
 mechanical grinding and acid-facilitated methods as a function of various oscillation
 frequencies, where the strain amplitude was controlled to 5%.

4) Page 7, the second passage in the Anchoring effect section:

The authors mention the advantage of the modified magnetic particles capped by Ag
 in terms of encapsulation of Fe₃O₄. However, in terms of durability, the alloying
 phenomenon between Ag and Indium could cause the excess alloy to deteriorate the
 property of magnetic LM composites, such as viscosity, or phase state (like the
 composite becomes turns into a solid state from a liquid state. Is there data about the
 time-dependence durability of the magnetic LM composite?

**Response:** Thanks a lot for your valuable comment. We totally agree with the
 reviewer that the excess alloying between Ag and In could deteriorate the fluidity of
 liquid metal composite, and thus the amount of Ag should be carefully controlled. In
 our work, an FPA (100 mM) mass fraction of 5% was selected to optimize the
 rheological property and magnetic responsiveness. After storage for an extended
 period, the durability data of EGaIn-5% FPA (100 mM) liquid metal composite in
 terms of rheological property were provided in **Fig. R2.3**. The phase of liquid metal
 composite was always at liquid state. Both the viscosity and moduli (storage modulus
 G' and loss modulus G'') kept relatively stable in the first 3 days, and then slightly
 increased after 7 days. Therefore, storage of EGaIn-5% FPA (100 mM) liquid metal
 composite in one week did not significantly affect its rheological property due to the
 limited production of intermetallic compounds in the liquid metal matrix. Such results
 have been added to the revised Supporting Information as Fig. S19.

 **Fig. R2.3.** (A) The viscosity of EGaIn-5% FPA (100 mM) composites as a function of
 shear rate after different periods. (B) The dynamic storage modulus and loss modulus
 of EGaIn-5% FPA (100 mM) composites as a function of oscillation frequency after
 different periods.

5) Page 8 and Figure 4, the first passage in the deformation section:

The authors mention that the fabricated sample with Fe₃O₄@PDA@Ag still has good
 viscosity even though the sample is a composite. However, will the fabricated sample
 via reactive wetting retain its viscosity over time? (an extension of the question about
 the excess alloying of Ag-In)

**Response:** Thanks for your comment. The time-dependent viscosity of EGaIn-5%
FPA (100 mM) has been provided in **Fig. R2.3**, indicating the fabricated composite
via reactive wetting was basically able to retain its viscosity over time.

**Fig. R2.4.** (A) Illustrations showing magnetic actuation cycle and the follow-up
deformation performance test for liquid metal composite. (B) The passive
deformation rate of liquid metal composite as a function of cycle number. (C) The
passive deformation rate of liquid metal composite as a function of storage time.

Except for viscosity, the deformation performance of the prepared liquid metal
composite over magnetic actuation cycle and time was also investigated. In the first
experiment, we used a permanent magnet to repeatedly drive EGaIn-5% FPA (100
mM) liquid metal composite-based LMMSR along a distance of 60 mm back and
forth with a speed of 10 cm/s in hydrochloric acid solution (1 M). After a certain
number of cycles (10, 50, 100, 150, and 200), the composite was transferred to a new
container to traverse a narrow slit via passive deformation under the guidance of a
permanent magnet (surface magnetic field ~ 500 mT) as shown in **Fig. R2.4A**. Then
**Fig. R2.4B** displayed the maximum deformation rate that the liquid metal composite
could achieve. It could be found that the deformation rate decreased from 75% to 70%
after 100 cycles, but the performance remained at 70% as the number of cycles
increased. These results indicated that the magnetic actuation performance of
LMMSR kept relatively stable and reliable after hundreds of actuation cycles. In the
second experiment, after the liquid metal composite was stored for different periods
(1, 3, and 7 days), its deformation performance was studied using the same setup. As

shown in **Fig R2.4C**, the storage time did not affect the deformation ability of liquid
metal composite, which should be attributed to the limited change of viscosity.

In summary, our fabricated EGaIn-5% FPA liquid metal composite via reactive
wetting could maintain its viscosity and magnetic actuation performance in one week.
Such stability could fully satisfy the demands of magnetic miniature soft robot in our
work for targeted cargo delivery, and the corresponding results have been added to the
revised Supporting Information as Fig. S22.

Response to Reviewer #3

A significant portion of the research article and the primary novelty focuses on
overcoming the intrinsic interfacial non-wettability between the Fe₃O₄ particles and
EGaIn using an intermediate silver layer via polydopamine to facilitate reactive
wetting and embed/anchor the Fe₃O₄ nanoparticles inside the EGaIn. However, this
strategy has already been introduced by other researchers such as Ref 35. While this
work is referenced by the authors, these prior contributions are largely ignored by the
authors and the scientific advancement over this work is unclear. The authors
contribution is largely incremental and not suitable for publication in Nature
Communications.

**Response:** Thanks for your comment. Despite the modification method of magnetic
agents, the material system, fundamental mechanism, as well as research objective
and significance of our work are all different from those of Ref 35 in the original
manuscript [*ACS Appl. Mater. Interfaces* 2021, 13, 5256-5265], which are
summarized in **Table R3.1**, and described in detail as follows.

**Material system.** In Ref 35, iron particles were coated with Ag shells, followed by
mixing with liquid metal. While in our work, iron oxide nanoparticles were
functionalized with Ag shells and then embedded in liquid metal. There existed an
essential difference between the pristine magnetic materials of Ref 35 and our work,
i.e., iron particles were intrinsically wettable with liquid metal, while in contrast, iron
oxide particles were non-wettable. We used inert and biocompatible iron oxide
particles rather than metallic magnetic particles (e.g., Fe, Ni, NdFeB, etc.) to
constitute magnetic liquid metal composite, because the potential leakage or
dissolution of metallic powders into surrounding environments may cause damage to
living organisms in potential biomedical applications.

**Fundamental mechanism.** Due to the distinct wettability between iron and iron oxide
particles with liquid metal, the fundamental working mechanism of silver
modification in our work becomes different from that in Ref 35. In the previous work,
Ag shells were coated on the surface of iron particles to serve as a sacrificial layer,
which could react with liquid metal and subsequently protect iron particles from
corrosion by liquid metal. Besides, the excellent electrical and thermal conductivities
of silver were utilized to improve the functionalities of liquid metal composite. In

sharp contrast, in our work, Ag shells were coated on the surface of non-wettable iron
oxide particles to serve as an intermediate layer between iron oxide and liquid metal,
which could react with liquid metal and significantly improve the wettability between
magnetic agents and liquid metal via reactive wetting mechanism. In summary, in Ref
35, silver modification was conducted to prevent the direct contact and wetting
between liquid metal and iron particles. While in our work, silver modification was
exploited to facilitate the contact and wetting of liquid metal on magnetic agents.

**Research objective.** In Ref 35, silver layer was coated to prevent the direct alloying
reaction between iron particles and liquid metal, or else transition in the crystal
structure of iron would occur, leading to the attenuation of magnetic responsiveness
of liquid metal composite. While in our work, the inert iron oxide particles did not
react with liquid metal at all, and moreover, were non-wettable with liquid metal. Our
aim of silver modification was to improve the wettability between iron oxide and
liquid metal via reactive wetting mechanism. In summary, the research objective of
Ref 35 was to obtain a magnetic liquid metal composite with good magnetism
stability, electroconductivity and thermoconductivity. While our research objective
was to embed and anchor iron oxide particles into liquid metal to obtain a magnetic
soft composite with satisfactory fluidity and biocompatibility.

**Research significance.** In Ref 35, a preparation strategy of iron particles/liquid metal
magnetic composite was proposed, which promoted the development of liquid metal
pastes with higher electrical and thermal conductivities for thermal management and
flexible electronics. In contrast, we reported a strategy to composite inert and
biocompatible iron oxide nanoparticles into EGaIn liquid metal via reactive wetting
mechanism, which could serve as a magnetic miniature soft robot to perform targeted
cargo delivery in an ex vivo porcine stomach. Therefore, the implemented
applications and target audiences between our work and Ref 35 are totally different.

Except for the difference mentioned above, we further investigated and revealed the
reaction competition between In and Ga with Ag elements, the dynamic wettability
transition between liquid metal and Ag modified magnetic agents, as well as the
medical imaging capability of liquid metal composite inside biological tissue. All of
these contents have not been studied before, which we believe constitutes a
nonnegligible scientific advancement that stimulates the development of soft robot in
biomedical fields, in particular towards in vivo or even clinic applications. In
summary, the modification strategy of Ag shell on the surface of magnetic agents is

similar, but there are fundamental differences between Ref 35 and our work. To avoid
 unnecessary misunderstanding, we have modified our statements, and the contribution
 of Ref 35 is described in Introduction section of the revised manuscript. We hope
 such response can address the concerns of reviewer at a satisfactory level, and
 convince the reviewer of the suitability of our work for publication in Nature
 Communications.

**Table R3.1.** Differences between Ref 35 of the original manuscript and our work.

	Ref 35 ACS Appl. Mater. Interfaces 2021, 13, 5256-5265	Our work
Material system	Iron particle, EGaIn liquid metal, polydopamine, silver *iron is intrinsically wettable with liquid metal.	Iron oxide particle, EGaIn liquid metal, polydopamine, silver *iron oxide is non-wettable with liquid metal.
Fundamental mechanism	 □ Ag shell serves as a sacrificial layer to protect iron particle from corrosion by liquid metal. □ Ag possesses excellent electrical and thermal properties. 	Ag shell serves as an intermediate layer between iron oxide and liquid metal to improve the wettability of liquid metal on magnetic agents via reactive wetting mechanism
Research objective	Improve the magnetism stability, electroconductivity and thermoconductivity of iron/liquid metal composite.	Prepare iron oxide/liquid metal composite with satisfactory fluidity and biocompatibility .
Research significance	Development of liquid metal pastes with higher electrical and thermal conductivities for thermal management and flexible electronics .	Development of magnetic miniature soft robots with satisfactory softness, magnetism and biosafety for interventional therapy and minimally invasive surgery .
Other contributions		 □ Reveal the reaction competition between In and Ga with Ag in liquid metal composite. □ Explore the dynamic wettability transition between liquid metal and Ag modified magnetic agents. □ Verify the medical imaging capability of iron oxide/liquid metal composite inside biological tissue.

434

REVIEWER COMMENTS

Reviewer #1 (Remarks to the Author):

The authors have significantly enhanced the quality of the manuscript and my concerns have been properly addressed. I recommend the publication of the manuscript.

Reviewer #2 (Remarks to the Author):

I thank the authors for their considerable efforts. The paper has improved.

There are still some lingering concerns that are more difficult to address:

1. Lack of novelty: The revised version of this paper actually showed that the novelty of the idea has decreased for the following two reasons:

- As stated in lines 75-77, the use of Ag as a sacrificial layer is a method that has already been used in other papers.
- The direct use of a magnet to move liquid metal is a commonly used method.
- As mentioned by Reviewer 3, a similar idea to this has already been used in Ref. 35.

2. Stability Issue: Although the authors claim that their liquid metal containing shell-modified Fe₃O₄ is stable, Movie 5 provided by the authors shows the metal is releasing powders, which are presumably Fe₃O₄ particles, as it deforms according to the magnet's shape. This seems to contradict the information presented in Figure S14? In S14, those samples were stable only in a static state within the solution, but if the magnetic particles separate from the liquid metal droplet containing Fe₃O₄ during the process of moving the droplet like in movie 5, it means that it is not stable, as claimed in this paper.

3. Questioning the significance of Figure S13: The figure provided in S13 indicates that Fe₃O₄ does not mix at all, but according to a paper published by Kong et al. in Adv. Mater., mixing can occur in the ambient environment by using oxide, especially in the case of tungsten, which cannot undergo

reactive wetting with liquid metal. Therefore, by adjusting the ratio of tungsten to liquid metal, the viscosity can also be adjusted. Therefore, it is difficult to see the claim provided in S13 as correct.

Reviewer #3 (Remarks to the Author):

The authors' response and revisions have satisfactorily addressed all reviewer comments on the previous version of the manuscript.

Reviewer #4 (Remarks to the Author):

This paper discusses a new strategy for creating magnetic liquid metal (LM) composites using iron oxide (Fe₃O₄) nanoparticles and eutectic gallium indium (EGaIn) LM. The authors introduce a silver intermediate layer to reduce compositional mismatch and improve wetting ability, resulting in significantly improved magnetic stability. They also construct a magnetic miniature soft robot that can perform controllable deformation and locomotion behaviors under external magnetic field actuation. Finally, they validate the practical feasibility of using the LM soft robot as a remotely controlled medical apparatus in an ex vivo porcine stomach.

However, there are major points which should be addressed:

The authors developed the bimetallic Ag-Fe₃O₄ which will be engulfed by EGaIn due to the high wetting behavior of Ag and GaLM. The main reason magnetic EGaIn can change shape is due to the various shapes of magnets. It means that it is quite similar to ferrofluid. The question is raised about the difference between these magnetic EGaIn and ferrofluid. There is no clear explanation of the advantage of using magnetic EGaIn instead of ferrofluid.

The authors claimed that Ag-Fe₃O₄ is inert with acid. However, they need to submerge these AgFe₃O₄ under acidic conditions of the stomach, then perform SEM/TEM/XRD to confirm their stability. Also, it is not clear how EGaIn remains stable under these acidic conditions.

Line 431-441, "Moreover, the leakage of metal elements (Ga, In, Fe, and Ag) from EGaIn-FPA composite was measured after immersing 500 mg EGaIn-5% FPA (20 mM) in 1 mL acidic solution (0.1M HCl). As shown in Fig. 6C, ..." In this text, the authors claimed the safe leakage or

biocompatibility. I think the authors should use these leakage and test them with common mammalian cells to see their viability, rather than assuming that they are not toxic, despite other studies.

A few other minor points:

Line 87, "Consequently, such oxide film.... the prepared functional composite" needs references.

The initial monodispersed Fe₃O₄ was ~850nm. Are they at a compatible size or bigger than the frequently used particle in the conventional method? After being fabricated, what is the size of the product (Fe₃O₄@FDA)?

Since the direct interaction between Ga-Ag did occur, is there any chance that the anchoring structure will be changed by time to become the fusion structure, in which the particle is fused with liquid metal?

Please consider changing the color of the line and the bar in the presented figure. The line/bar color is too light! They need to be adjusted to a darker tone (i.e., the cyan blue/pink can be switched to blue/red, the light grey line can be changed to dark gray) for better visualization.

Response to Reviewer #1

The authors have significantly enhanced the quality of the manuscript and my concerns have been properly addressed. I recommend the publication of the manuscript.

Response: Thank you very much for your valuable comments, which we believe help to significantly improve our manuscript.

Response to Reviewer #2

I thank the authors for their considerable efforts. The paper has improved. There are still some lingering concerns that are more difficult to address:

1. Lack of novelty: The revised version of this paper actually showed that the novelty of the idea has decreased for the following two reasons:

- As stated in lines 75-77, the use of Ag as a sacrificial layer is a method that has already been used in other papers.

Response: Thanks for your comment. In the previous work [*ACS Appl. Mater. Interfaces* 2021, 13, 5256] (cited in our revised manuscript as Ref. [41]), Ag was coated on iron particles to prevent the direct contact and wetting between liquid metal and iron particles, aiming to protect iron particles from corrosion. While in our work, Ag was coated on iron oxide particles to promote the contact and wetting of liquid metal on iron oxide, aiming to composite non-wettable particles into liquid metal. Therefore, the fundamental working mechanism of Ag sacrificial layer in our work is completely opposite to that in previous work. There exist substantial differences in material system, fundamental mechanism, research objective and significance. Here the differences are emphasized and summarized in **Table R2.1** for your reference.

Material system. In Ref. [41], iron particles were coated with Ag shells, followed by mixing with liquid metal. While in our work, iron oxide nanoparticles were functionalized with Ag shells and then embedded in liquid metal. There existed an essential difference between the pristine magnetic materials of Ref. [41] and our work, i.e., iron particles were intrinsically wettable with liquid metal, while in contrast, iron oxide particles were non-wettable. We used inert and biocompatible iron oxide particles rather than metallic magnetic particles (e.g., Fe, Ni, NdFeB, etc.) to constitute magnetic liquid metal composite, because the potential leakage or dissolution of metallic powders into surrounding environments may cause damage to living organisms in biomedical applications.

Fundamental mechanism. Due to the distinct wettability between iron and iron oxide particles with liquid metal, the fundamental working mechanism of silver modification in our work becomes different from that in Ref. [41]. In the previous work, Ag shells were coated on the surface of iron particles to serve as a sacrificial layer, which could react with liquid metal and subsequently protect iron particles from

corrosion by liquid metal. Besides, the excellent electrical and thermal conductivities of silver were utilized to improve the functionalities of liquid metal composite. In sharp contrast, in our work, Ag shells were coated on the surface of non-wettable iron oxide particles to serve as an intermediate layer between iron oxide and liquid metal, which could react with liquid metal and significantly improve the wettability between magnetic agents and liquid metal via reactive wetting mechanism. In summary, in Ref. [41], silver modification was conducted to prevent the direct contact and wetting between liquid metal and iron particles. While in our work, silver modification was exploited to facilitate the contact and wetting of liquid metal on magnetic agents.

Research objective. In Ref. [41], silver layer was coated to prevent the corrosion of iron particles by liquid metal, or else transition in the crystal structure of iron would occur, leading to the attenuation of magnetic responsiveness of liquid metal composite. While in our work, the inert iron oxide particles did not react with liquid metal at all, and moreover, were non-wettable with liquid metal. Our aim of silver modification was to improve the wettability between iron oxide and liquid metal via reactive wetting mechanism. In summary, the research objective of Ref. [41] was to obtain a magnetic liquid metal composite with good magnetism stability, electroconductivity and thermoconductivity. While our research objective was just to embed and anchor iron oxide particles into liquid metal to obtain a magnetic soft composite with satisfactory fluidity and biocompatibility.

Research significance. In Ref. [41], a preparation strategy of iron particles/liquid metal magnetic composite was proposed, which promoted the development of liquid metal pastes with higher electrical and thermal conductivities for thermal management and flexible electronics. In contrast, we reported a strategy to composite inert and biocompatible iron oxide nanoparticles into EGaln liquid metal via reactive wetting mechanism, which could serve as a magnetic miniature soft robot to perform targeted cargo delivery in an ex vivo porcine stomach. Therefore, the implemented applications and target audiences between our work and Ref. [41] are totally different.

Except for the differences mentioned above, we further investigated and revealed the reaction competition between In and Ga with Ag elements, the dynamic wettability transition between liquid metal and Ag modified magnetic agents, as well as the medical imaging capability of liquid metal composite inside biological tissue. All of these contents have not been studied before, which we believe constitute a nonnegligible scientific advancement that stimulates the development of soft robot in biomedical fields, in particular towards in vivo or even clinic applications.

Table. R2.1. Difference between Ref. [41] and our work.

	Ref [41] ACS Appl. Mater. Interfaces 2021, 13, 5256	Our work
Material system	Iron particle, EGaln liquid metal, polydopamine, silver *iron is intrinsically wettable with liquid metal.	Iron oxide particle, EGaln liquid metal, polydopamine, silver *iron oxide is non-wettable with liquid metal.
Fundamental mechanism	 □ Ag shell serves as a sacrificial layer to protect iron particle from corrosion by liquid metal. □ Ag possesses excellent electrical and thermal properties. 	Ag shell serves as an intermediate layer between iron oxide and liquid metal to improve the wettability of liquid metal on magnetic agents via reactive wetting mechanism
Research objective	Improve the magnetism stability, electroconductivity and thermoconductivity of iron/liquid metal composite.	Prepare iron oxide/liquid metal composite with satisfactory fluidity and biocompatibility .
Research significance	Development of liquid metal pastes with higher electrical and thermal conductivities for thermal management and flexible electronics .	Development of magnetic miniature soft robots with satisfactory softness, magnetism and biosafety for interventional therapy and minimally invasive surgery .
Other contributions		 □ Reveal the reaction competition between In and Ga with Ag in liquid metal composite. □ Explore the dynamic wettability transition between liquid metal and Ag modified magnetic agents. □ Verify the medical imaging capability of iron oxide/liquid metal composite inside biological tissue.

In summary, the usage of Ag as a sacrificial layer is similar, but there are fundamental differences between Ref. [41] and our work. We hope such response can address the concerns of reviewer at a satisfactory level, and convince the reviewer of the novelty and suitability of our work for publication in Nature Communications.

- The direct use of a magnet to move liquid metal is a commonly used method.

Response: Thanks for your comment. The main contribution of our work is to composite non-wettable, inert and biocompatible Fe₃O₄ magnetic nanoparticles into EGaln liquid metal via reactive wetting mechanism for the development of magnetic miniature soft robot. The use of magnet just serves as a tool to generate different magnetic fields, rather than constitutes the key novelty of our work. We do not think it weakens the quality and significance of work. On the other hand, using a common method to successfully control our newly developed liquid metal soft robot can promote the realization of practical applications, as there is no need to develop new control methods. Actually, there are mainly two magnetic control methods, that is, permanent magnet and electromagnet. A variety of ferrofluid-based soft robots have been developed and controlled by these two kinds of magnets. Some of them are also actuated by permanent magnets, and have been recently published in top journals, such as [*Sci. Adv.* 2022, 8, eabq1677], [*PNAS* 2020, 117, 27916], and [*Nature* 2018, 559, 77]. Therefore, although we use a common method to control the magnetic liquid metal composite, we believe the novelty and significance of our work make it deserve the consideration for publication in Nature Communications.

- As mentioned by Reviewer 3, a similar idea to this has already been used in Ref. 35.

Response: Thanks for your comment. We have provided a detailed explanation on the differences between our work and the previous paper (**Ref. [35] in the previous manuscript, which is Ref. [41] in the revised manuscript**) in the response to Comment 1. We hope our statement can convince the reviewer of the novelty of our work. Reviewer 3 is satisfactory with our response and recommends the publication of our work.

2. Stability Issue: Although the authors claim that their liquid metal containing shell-modified Fe_3O_4 is stable, Movie 5 provided by the authors shows the metal is releasing powders, which are presumably Fe_3O_4 particles, as it deforms according to the magnet's shape. This seems to contradict the information presented in Figure S14? In S14, those samples were stable only in a static state within the solution, but if the magnetic particles separate from the liquid metal droplet containing Fe_3O_4 during the process of moving the droplet like in movie 5, it means that it is not stable, as claimed in this paper.

Response: Thanks for your comment. In **previous Movie S5**, the magnetic liquid metal composite was releasing bubbles, rather than Fe_3O_4 nanoparticles. The solution used was 1 M HCl solution, so reaction between liquid metal and acid occurred to produce abundant hydrogen bubbles. We used such high concentration of acid to completely remove the oxide film on the surface of magnetic liquid metal composite, making the deformation of composite under different magnetic fields more rapid and robust. While **Fig. S14 in the previous Supporting Information, which is Fig. S15 in the revised Supporting Information**, characterized the liquid metal composite in HCl solution with a concentration of 0.3 M, demonstrating the good stability of magnetic particles in EGaIn. **Movie S5** and **Fig. S15** were not contradictory as the concentration of HCl in solution was different. With the decrease of HCl concentration, liquid metal composite became increasingly stable, especially when the concentration was lower than 0.5 M, as shown in **Fig. R2.1**. Other experimental results in the Supporting Information demonstrated that the magnetic liquid metal composite could exhibit good deformability in HCl solution with a concentration of $\sim 0.1\text{-}0.2$ M, which fully satisfied the application demands of magnetic miniature soft robot in stomach (HCl concentration in gastric acid is ~ 0.1 M).

In order to avoid confusion, we remade a video that demonstrated the shape switching

and reconfiguration of liquid metal-based magnetic soft robot as **new Movie S5**. At this time, the generation of bubbles was, to a great extent, suppressed by manipulating liquid metal soft robot in acidic solution with relatively low HCl concentration (~ 0.5 M). **Fig. R2.2** also presented some snapshots of **new Movie S5**, in which, the liquid metal soft robot still rapidly switched its morphology between a circle and a triangle under the actuation of external magnetic fields.

Fig. R2.1. Optical images showing the stability of magnetic liquid metal composites in HCl solution with different concentrations.

Fig. R2.2. Sequential video snapshots showing the deformation of liquid metal-based magnetic soft robot under different external magnetic fields.

At last, we would like to state that our developed magnetic EGaln liquid metal composite was not completely stable in simulated gastric acid (i.e., ~ 0.1 M HCl solution). After immersing 500 mg composite in 1 mL 0.1 M HCl solution for 1 h (the time was long enough for targeted cargo transportation in stomach), the concentration of leaked Ga, In, Fe and Ag in solution was measured to be 17.40, 1.90, 33.52, and 3.94 mg/L, respectively (**Fig. R2.3A**). Only ~ 17.4 μ g Ga and ~ 1.9 μ g In could be

detected in the acid after 1 h, indicating EGaIn liquid metal was consumed very slowly. Such limited consumption would not attenuate the actuation and deformation performance of magnetic liquid metal composite, as shown in the Supporting Information. We also investigated whether these leaked metal ions were toxic to mammalian cells by reproducing the concentration of each metal element in cell culture medium. The results in **Fig. R2.3B** indicated that human embryonic kidney HEK293 cells maintained excellent viability (>90%) even after 6 h of incubation with the leaked metal ions. In summary, our magnetic liquid metal composite could keep almost stable in both dynamic and static states, when the HCl concentration of acidic solution was close to the gastric acid in stomach. A very limited amount of liquid metal would be dissolved by acid after 1 h, and the leaked ions were not harmful to mammalian cells. All the data were consistent in our work.

Fig. R2.3. (A) The concentrations of heavy metals measured in the supernatant of HCl solution (0.1 M) containing magnetic liquid metal composite over time. Error bars indicate the SD for $n = 3$. (B) Viability of HEK293 cells co-cultured with the medium containing Ga (17.40 mg/L), In (1.90 mg/L), Fe (33.52 mg/L) and Ag (3.94 mg/L). Error bars indicate the SD for $n = 3$.

3. Questioning the significance of Figure S13: The figure provided in S13 indicates that Fe_3O_4 does not mix at all, but according to a paper published by Kong et al. in *Adv. Mater.*, mixing can occur in the ambient environment by using oxide, especially in the case of tungsten, which cannot undergo reactive wetting with liquid metal. Therefore, by adjusting the ratio of tungsten to liquid metal, the viscosity can also be adjusted. Therefore, it is difficult to see the claim provided in S13 as correct.

Response: Thanks for your comment. In the **Fig. S13 of previous Supporting Information, which is Fig. S14 of the revised Supporting Information**, we would like to emphasize that the mechanical grinding process was performed in a glovebox filled with nitrogen gas. There was no oxygen in the environment during grinding.

The results were also provided here as **Fig. R2.4** for your reference. At this time, Fe_3O_4 nanoparticles did not mix with liquid metal at all in nitrogen atmosphere. While in the work suggested by the reviewer [*Adv. Mater.* 2019, 31, 1904309], tungsten powders were also unable to wet with liquid metal in a nitrogen environment (**Fig. R2.5**), which was consistent with our results. Successful mixing could only occur in the ambient environment where oxygen existed.

Fig. R2.4. Mechanical grinding of EGaln liquid metal with FPA (A) and Fe_3O_4 (B) powders in a glovebox filled with nitrogen gas.

Fig. R2.5. Results from the paper [*Adv. Mater.* 2019, 31, 1904309] suggested by reviewer. When mixed in nitrogen atmosphere, tungsten powders cannot mix with liquid metal at all (a), which is consistent with the results in our work. In the ambient environment with enough oxygen, tungsten powders become able to mix with liquid metal (c), due to the formation of a large amount of liquid metal oxides wrapped on tungsten particles. Ag particles can always be composited into liquid metal in N_2 atmosphere (b) and air (d) due to reactive wetting mechanism.

Actually, other non-wettable particles that could not directly react with liquid metal, such as graphene oxide, graphite, diamond, and silicon carbide, were also reported to be capable of mixing with liquid metal when there was enough oxygen to produce a lot of liquid metal oxides [*Sci. Adv.* 2021, 7, eabe3767]. But it was a time-consuming and labor-intensive procedure that required the production of liquid metal oxide films to wrap on doping particles. Consequently, such oxide films would attenuate the fluidity of liquid metal, yet if they were eliminated, the doped particles would easily leak out under external stimulus, leading to the invalidation of prepared functional liquid metal composites. Using oxides to prepare magnetic liquid metal composite is not suitable for magnetic miniature soft robot, as it cannot guarantee satisfactory softness, magnetism and stability for robotic applications. Therefore, we develop a facile preparation method of magnetic liquid metal composite via reactive wetting between liquid metal and Ag shell modified Fe₃O₄ nanoparticles, aiming to effectively incorporate non-wettable iron oxide into liquid metal while not deteriorating the fluidity of liquid metal.

In summary, our experimental results are not contradictory with the previous paper published by Kong et al., which indicated Fe₃O₄ or tungsten particles cannot mix with liquid metal at all when there was no oxygen in the environment.

Response to Reviewer #3

The authors' response and revisions have satisfactorily addressed all reviewer comments on the previous version of the manuscript.

Response: We are pleased that our revisions have satisfactorily addressed your concerns. Thank you very much for your valuable comments, which help us to refine our work and enhance the quality of manuscript.

Response to Reviewer #4

This paper discusses a new strategy for creating magnetic liquid metal (LM) composites using iron oxide (Fe_3O_4) nanoparticles and eutectic gallium indium (EGaIn) LM. The authors introduce a silver intermediate layer to reduce compositional mismatch and improve wetting ability, resulting in significantly improved magnetic stability. They also construct a magnetic miniature soft robot that can perform controllable deformation and locomotion behaviors under external magnetic field actuation. Finally, they validate the practical feasibility of using the LM soft robot as a remotely controlled medical apparatus in an ex vivo porcine stomach. However, there are major points which should be addressed.

Response: Thanks a lot for your comprehensive summary. We have answered every comment and made corresponding changes in the manuscript. Please check the following point-by-point response for details.

The authors developed the bimetallic Ag- Fe_3O_4 which will be engulfed by EGaIn due to the high wetting behavior of Ag and Ga LM. The main reason magnetic EGaIn can change shape is due to the various shapes of magnets. It means that it is quite similar to ferrofluid. The question is raised about the difference between these magnetic EGaIn and ferrofluid. There is no clear explanation of the advantage of using magnetic EGaIn instead of ferrofluid.

Response: Thanks for your comment. As the reviewer stated, we agree that our magnetic LM composite actually belongs to the category of liquid-based magnetic materials. However, compared to the well-known ferrofluids (i.e., magnetically responsive fluid using water or organic solvents as substrate), magnetic EGaIn LM composite developed in this work exhibits three notable advantages for biomedical applications, including appealing immiscibility, biocompatibility and medical imaging capability, which are described as follows.

To be used for biomedical applications, fluid-based magnetic soft robot should firstly be immiscible in complex physiological environments, or else it will easily diffuse into biofluids and/or adhere to the tissue surface, which not only lead to the invalidation of functions, but also cause potential biosafety risks. Herein, we purchased three types of conventional ferrofluids, i.e., mineral oil-based, alkane-based, and water-based ferrofluids, and compared their immiscibility with that of our

magnetic EGaIn LM composite. 500 μL of each magnetic fluid was added to an ex vivo porcine stomach filled with simulated gastric acid (0.1 M HCl solution), and then removed with a pipette after ~ 1 min. The surface of stomach wall was recorded in **Fig. R4.1**. It could be found that magnetic EGaIn LM composite (Fig. R4.1A) did not adhere to the stomach wall and left no residual material upon removal. In contrast, both mineral oil-based and alkane-based ferrofluids in Fig. R4.1B & C adhered to the stomach wall due to their polar similarity to gastric mucosa, and the corresponding residual materials could not be removed even being flushed by phosphate buffer saline (PBS). As to the water-based ferrofluid in Fig. R4.1D, it rapidly diffused into the simulated gastric acid upon addition, which could not be actuated by magnetic field or removed by a pipette. Therefore, compared to the commercially available mineral oil-based, alkane-based, and water-based ferrofluids, liquid metal-based magnetic composite exhibits superior immiscibility, which benefits the implementation of biomedical applications.

Fig. R4.1. Optical images showing the appearance of stomach after addition (left) and removal (right) of EGaIn-FPA magnetic LM composite (A), mineral oil-based ferrofluid (B), alkane-based ferrofluid (C), and water-based ferrofluid (D). The insets showed the magnified areas surrounded by red boxes.

Besides, another prerequisite for the soft robots used in biomedical area is that the material must be biocompatible. Herein, magnetic EGaIn LM composite, mineral oil-

based and alkane-based ferrofluids were cocultured with human embryonic kidney (HEK 293) cells. After 24 h, a nonradioactive colorimetric Cell Counting Kit-8 (CCK-8) assay was carried out to quantitatively evaluate the viabilities of cells. As shown in **Fig. R4.2**, when the concentration of magnetic material was as low as 12 $\mu\text{g/mL}$, all the tested fluids were not harmful to HEK293 cells. However, with the increase of concentration, the cell viabilities experienced a sharp decrease for mineral oil-based and alkane-based ferrofluids. In contrast, more than 80% of cells were still alive when they were incubated with magnetic EGaIn LM composite even with a high concentration of 200 $\mu\text{g/mL}$. Therefore, our developed LM-based magnetic composite exhibits superior biocompatibility than mineral oil-based and alkane-based ferrofluids.

Fig. R4.2. Cell viability of HEK293 cells cultured with EGaIn-FPA magnetic LM composite, mineral oil-based ferrofluid, and alkane-based ferrofluid at various concentrations for 24 h. Error bars indicate the SD for $n = 6$.

As liquid metal is opaque to X ray, our developed magnetic EGaIn LM composite may exhibit stronger X ray-based image contrast than other commercially available ferrofluids, contributing to the better control of soft robot inside biological tissue using medical imaging feedback. This was verified via an irradiator system. As shown in **Fig. R4.3A**, six kinds of liquid, including pure water, water-based ferrofluid, mineral oil-based ferrofluid, alkane-based ferrofluid, magnetic EGaIn LM composite, and pure EGaIn LM, were added to a 24-well plate and imaged using X ray. The fluids containing LM possessed much stronger image contrast compared to other fluids. Even when a fresh porcine stomach was placed over the 24-well plate (**Fig. R4.3B**), the groups containing LM still distinguished themselves from the biological

tissues, while the other fluids were not visible compared to background signal. Therefore, LM-based magnetic composite possesses better biomedical imaging performance for promising in vivo applications than other conventional ferrofluids.

Fig. R4.3. (A) Optical image and CT image for various fluids. (B) Optical image and CT image for various fluids after covering porcine stomach.

In summary, the advantages of magnetic EGaIn LM composite over conventional ferrofluids include superior immiscibility with biological environment, lower toxicity to normal cells and distinguished medical imaging capability, making the magnetic LM composite become a promising material for the development of medical miniature soft robot.

The authors claimed that $\text{Ag-Fe}_3\text{O}_4$ is inert with acid. However, they need to submerge these $\text{Ag-Fe}_3\text{O}_4$ under acidic conditions of the stomach, then perform SEM/TEM/XRD to confirm their stability. Also, it is not clear how EGaIn remains stable under these acidic conditions.

Response: Thanks for your comments. According to your suggestions, $\text{Fe}_3\text{O}_4@\text{PDA}@\text{Ag}$ (FPA) nanoparticles were dispersed in simulated gastric acid (0.1M HCl solution) for 2 h and then washed with deionized water, followed by XRD, SEM and TEM characterizations to evaluate their stability. Detailed results are shown as follows.

Fig. R4.4A showed that there was no significant change for FPA nanoparticles before and after gastric acid treatment. Strong diffraction peaks corresponding to Fe_3O_4 and Ag remained almost same after 2 h. Through zooming into the XRD pattern (**Fig. R4.4B**) of acid-treated FPA nanoparticles, new yet weak peaks corresponding to AgCl were detected, suggesting the reaction between Ag on the nanoparticle surface and acid. However, such peaks only accounted for a negligible portion of the whole XRD pattern, demonstrating that the FPA nanoparticles could keep stable during 2h incubation in gastric acid.

Fig. R4.4. (A) XRD analysis for FPA nanoparticles before and after simulated gastric acid treatment. The characteristic peaks for Fe_3O_4 , AgCl, and Ag are provided for reference. (B) Magnified XRD pattern of FPA nanoparticles after acid treatment.

Fig. R4.5. (A) SEM image of FPA nanoparticles before acid treatment. (B) SEM image of FPA nanoparticles after acid treatment. (C) SEM image and coupled EDX spot analysis of FPA nanoparticle after acid treatment.

Besides, **Fig. R4.5A** and **Fig. R4.5B** showed the morphologies of FPA nanoparticles using SEM before and after acid treatment, respectively. Both kinds of particles exhibited Ag particles or shells with similar morphologies on the outside surfaces, regardless of whether they were dispersed in acid or not. Spot scanning data of EDX

analysis in **Fig. R4.5C** revealed the presence of Ag, Fe, C, and O elements on the surface of acid-treated FPA nanoparticle (Si element came from the silicon wafer substrate), confirming the stability of FPA particles during 2h incubation in acidic environment.

Similarly, TEM images and corresponding EDX mapping analysis of the FPA nanoparticles before and after acid treatment (**Fig. R4.6** and **Fig. R4.7**) also showed no significant change. **Fig. R4.8** further provided the enlarged surface morphology of acid-treated FPA nanoparticle. It could be found that Ag particles still adhered to the nanoparticle surface with an intact interface. No obvious corrosion or dissolution of the Ag particles and inner Fe_3O_4 nanoparticle occurred under acidic conditions. In summary, the XRD, SEM, and TEM results verify that FPA nanoparticles are relatively inert with simulated gastric acid, paving the foundation for next-step biomedical applications in stomach.

Fig. R4.6. TEM image and EDX mappings of FPA nanoparticles before acid treatment.

Fig. R4.7. TEM image and EDX mappings of FPA nanoparticles after acid treatment.

Fig. R4.8. Magnified TEM images of FPA nanoparticle after acid treatment.

As to the EGaIn LM, we totally agree that it will become unstable in highly acidic environment. We have immersed the magnetic EGaIn LM composite in HCl solution with different concentrations. As shown in **Fig. R4.9A**, when the concentration of HCl solution was 1 M, EGaIn LM was corroded, accompanied by the constant production of hydrogen bubbles on the surface. However, with the decrease of HCl concentration, magnetic EGaIn LM composite became increasingly stable, especially when HCl concentration was lower than 0.2 M, which fully satisfied the application demands of magnetic miniature soft robot in stomach (HCl concentration in gastric acid was ~ 0.1 M). Besides, it was reported that when immersing liquid metal in acidic solution, a HCl concentration lower than 0.2 M was insufficient to thoroughly prevent the oxidation of liquid metal [*Phys. Fluids* 2012, 24, 063101]. Therefore, we believed the good stability of magnetic EGaIn LM composite in simulated gastric acid was mainly due to the formation of a thin oxide film on the surface, which effectively inhibited the corrosion of liquid metal by acid.

Actually, we would like to state that EGaIn LM was not completely stable. We immersed 500 mg magnetic EGaIn LM composite in 1 mL simulated gastric acid and measured the leakage of metal elements from the composite after different time. As shown in **Fig. R4.9B**, although the concentration of Ga and In slowly increased as time went on, only ~ 17.4 μg Ga and ~ 1.9 μg In could be detected in the acid after 1 h, indicating EGaIn was consumed very slowly.

In summary, both magnetic particles and EGaIn LM can keep almost stable when the HCl concentration of acidic solution is close to the gastric acid in stomach. A very limited amount of composite would be dissolved by acid after 1 h, and the leaked ions are not harmful to mammalian cells, which is verified in the response to the next comment.

Fig. R4.9. Stability of magnetic EGaIn LM composite in HCl solution. (A) Optical images of magnetic EGaIn LM composite in HCl solution with different concentrations. (B) The concentrations of leaked metal elements measured in the supernatant of 0.1 M HCl solution containing magnetic EGaIn LM composite over time. Error bars indicate the SD for n = 3.

Line 431-441, "Moreover, the leakage of metal elements (Ga, In, Fe, and Ag) from EGaIn-FPA composite was measured after immersing 500 mg EGaIn-5% FPA (20 mM) in 1 mL acidic solution (0.1M HCl). As shown in Fig. 6C, ..." In this text, the authors claimed the safe leakage or biocompatibility. I think the authors should use this leakage and test them with common mammalian cells to see their viability, rather than assuming that they are not toxic, despite other studies.

Response: Thanks for your suggestion. We have provided supplementary experiments about the biocompatibility of leaked metal elements (Ga, In, Fe, and Ag). Detailed experiment process and data are shown as follows.

After immersing 500 mg magnetic EGaIn LM composite in 1 mL 0.1 M HCl solution for 1 h (the time was long enough for targeted cargo transportation), the concentration of leaked Ga, In, Fe and Ag was 17.40, 1.90, 33.52, and 3.94 mg/L, respectively (**Fig. R4.9B**). Then we dissolved anhydrous FeCl₃, GaCl, InCl₃, and AgNO₃ powders in cell culture medium to reproduce the concentration of each metal element, aiming to investigate whether these leaked metal ions were toxic to common mammalian cells. Human embryonic kidney HEK293 cells were co-cultured with the prepared medium in 96-well plate for different times (2, 4, 6 and 12 h). Then a nonradioactive colorimetric CCK-8 assay was carried out to quantitatively evaluate the viabilities of cells. As shown in **Fig. R4.10**, HEK293 cells maintained excellent viability (>90%) after 6 h of incubation with the leaked metal ions. Further increasing the co-culture time to 12 h only led to a slight decrease of cell viability to ~80.7%. It should be

noted that the volume of gastric acid in practical scenario was much larger than 1 mL, so the real concentration of leaked metal ions should be lower than the tested concentration due to the diffusion of ions. In this manner, cell viability would be higher. Therefore, it could be concluded that manipulating magnetic EGaIn LM composite in stomach within 1 h was not harmful to mammalian cells.

Fig. R4.10. Viability of HEK293 cells co-cultured with the medium containing Ga (17.40 mg/L), In (1.90 mg/L), Fe (33.52 mg/L) and Ag (3.94 mg/L). Error bars indicate the SD for n = 3.

A few other minor points:

Line 87, “Consequently, such oxide film.... the prepared functional composite” needs references.

Response: Thanks for your comment. According to your suggestion, we have cited relevant papers in the revised manuscript as Ref. [47] and [48].

The initial monodispersed Fe_3O_4 was ~850 nm. Are they at a compatible size or bigger than the frequently used particle in the conventional method? After being fabricated, what is the size of the product ($\text{Fe}_3\text{O}_4@FDA$)?

Response: Thanks for your comment. The Fe_3O_4 particles used in our work were fabricated by hydrothermal method, and their average size is close to that of the previous paper [*Angew. Chem. Int. Ed.* 2005, 44, 2782]. However, when compared to the magnetic agents in conventional ferrofluids (average size lower than 100 nm), our magnetic particles are indeed bigger.

Through measuring and summarizing the diameters of Fe_3O_4 and $\text{Fe}_3\text{O}_4@\text{PDA}$ nanoparticles in SEM images (**Fig. R4.11**), the size of $\text{Fe}_3\text{O}_4@\text{PDA}$ particles is 960 ± 120 nm, which is larger than that of Fe_3O_4 particles.

Fig. R4.11. (A) SEM image showing monodispersed Fe_3O_4 nanoparticles. (B) SEM image showing fabricated $\text{Fe}_3\text{O}_4@\text{PDA}$ nanoparticles. (C) Size comparison between Fe_3O_4 and $\text{Fe}_3\text{O}_4@\text{PDA}$. The data are obtained by measuring the diameters of 100 nanoparticles in the SEM images.

Since the direct interaction between Ga-Ag did occur, is there any chance that the anchoring structure will be changed by time to become the fusion structure, in which the particle is fused with liquid metal?

Response: Thanks for your comment. In our experiments, we did not find that the anchoring structure became fusion structure, or the particles were fused with liquid metal. After compositing FPA nanoparticles into EGaIn LM and preserving the composite for over one week, micro-CT was employed to study the phase and structure inside liquid metal. **Fig. R4.12** clearly presented the existence of several high-density Ag_xIn_y phases, indicating that the anchoring structure did not become fusion structure within one week. We further emulsified the magnetic EGaIn LM composite into microparticles, and used TEM to characterize the composition and morphology of composite. From the elemental distribution in **Fig. R4.13**, there was a clear boundary between Fe (O) and Ga elements, indicating Fe_3O_4 nanoparticles were not inclined to be fused with liquid metal. The overlap between Ag and In elements suggested the formation of anchoring phase Ag_xIn_y .

Fig. R4.12. Micro-CT of magnetic EGaIn LM composite, where the 3D image (left) shows the distribution of high-density Ag_xIn_y phase (green regions) in LM (cylindrical dotted region), the slice image (middle) exhibits the horizontal section of composite, and the illustration in the right corner indicates the anchoring mechanism between FPA and EGaIn LM.

Fig. R4.13. TEM image and corresponding EDX mapping analysis of magnetic EGaIn LM composite-based microparticles, indicating Fe_3O_4 nanoparticles are not fused with liquid metal.

Please consider changing the color of the line and the bar in the presented figure. The line/bar color is too light! They need to be adjusted to a darker tone (i.e., the cyan blue/pink can be switched to blue/red, the light grey line can be changed to dark gray) for better visualization.

Response: Thanks for your suggestion. We have adjusted the line and bar colors of figures in the revised manuscript. We hope such revision can address your concern at a satisfactory level.

REVIEWER COMMENTS

Reviewer #2 (Remarks to the Author):

Thank you to the authors for addressing the prior concerns. Of the prior questions/ concerns, the first and third were addressed satisfactorily, but I have some questions about the second.

Response 1. This response is acceptable as the authors explained the novelty by focusing on differences from the previous research in terms of 'Material system, fundamental mechanism, research objective, and research significance'.

Response 2. In the first paragraph of the response, the authors seem to imply that the component that separated from LM in movie 5 was not F3O4 powder but bubbles, suggesting that their composite is stable in acid. Ag does not react with HCl, so the bubbles came from the reaction between liquid metal and HCl.

However, I wonder if this claim is contradictory in the following sense:

-The authors claim that the movie shows bubbles, not powder, so there should be no issue with powder encapsulation, and their composite is stable in acid. However, if the liquid metal composite generated bubbles in acid, it means that the acid penetrated several layers (Fe3O4 coating layer, Ag sacrificial layer) and interacted with LM, causing bubble formation. At this point, bubbles formed on the LM surface could physically damage the Ag layer and further result in the detachment of powder from LM.

- Moreover, the authors mentioned that they reduced the concentration of HCl after describing the above scenario. This implies that controlling the acid's concentration is necessary, indicating that their composite is not stable in acid. However, they do provide evidence that the composite could be relatively stable in the acid found in the stomach.

Response 3. Response is acceptable.

Reviewer #4 (Remarks to the Author):

The authors have addressed all of my concerns. It should be accepted for publication.

Response to Reviewer #2

Thank you to the authors for addressing the prior concerns. Of the prior questions/concerns, the first and third were addressed satisfactorily, but I have some questions about the second

Response 1. This response is acceptable as the authors explained the novelty by focusing on differences from the previous research in terms of 'Material system, fundamental mechanism, research objective, and research significance'.

Response: Thanks a lot for your recognition of our work. We have answered every comment, and please check the following point-by-point response for details.

Response 2. In the first paragraph of the response, the authors seem to imply that the component that separated from LM in movie 5 was not Fe_3O_4 powder but bubbles, suggesting that their composite is stable in acid. Ag does not react with HCl, so the bubbles came from the reaction between liquid metal and HCl.

However, I wonder if this claim is contradictory in the following sense:

-The authors claim that the movie shows bubbles, not powder, so there should be no issue with powder encapsulation, and their composite is stable in acid. However, if the liquid metal composite generated bubbles in acid, it means that the acid penetrated several layers (Fe_3O_4 coating layer, Ag sacrificial layer) and interacted with LM, causing bubble formation. At this point, bubbles formed on the LM surface could physically damage the Ag layer and further result in the detachment of powder from LM.

Response: Thanks for your comment. To avoid misunderstanding, we would like to describe the preparation process of magnetic EGaIn LM composite (**Fig. R2.1A**). Firstly, magnetic agents $\text{Fe}_3\text{O}_4@\text{PDA}@\text{Ag}$ (FPA) nanoparticles were prepared by coating polydopamine (PDA) and silver layers on the surface of Fe_3O_4 nanoparticles in sequence. Then with the assistance of Ag sacrificial layer, FPA nanoparticles were mixed and composited with EGaIn LM via reactive wetting mechanism, while the Ag layer changed to Ag_xIn_y intermetallic compound (IMC). At this time, FPA nanoparticles were dispersed among the matrix of LM, including both the surface and interior of LM. This could be verified by the micro-CT characterization of magnetic EGaIn LM composite. As shown in **Fig. R2.1B** and **Movie S4**, the cylindrical dotted

region represented LM phase, while the green phases were high-density Ag_xIn_y IMCs, indicating that Ag_xIn_y IMCs distributed uniformly among LM. Through further horizontally slicing the micro-CT image, Fe_3O_4 nanoparticles coated with Ag_xIn_y IMCs could be detected, and Ag_xIn_y IMCs served as the anchoring sites to stably embed magnetic agents into LM. Therefore, when magnetic EGaIn LM composite was immersed in highly acidic solution (e.g., 1 M HCl solution), acid would first encounter the surface LM oxide film of composite and dissolve the LM oxides. Afterwards, EGaIn LM and Ag_xIn_y IMCs of the surface distributed FPA nanoparticles were exposed to acidic solution (**Fig. R2.1C**), which would form a galvanic cell to produce hydrogen bubbles, similar to that reported in [*Mater. Horiz.* 2018, 5, 222].

Fig. R2.1. (A) Preparation scheme of magnetic EGaIn LM composite. (B) Micro-CT analysis of magnetic EGaIn LM composite. (C) Scheme of bubble formation when immersing magnetic EGaIn LM composite in highly acidic solution.

More specifically, for the formed EGaIn-Ag_xIn_y electrode system in HCl solution, as the standard electrode potentials of Ga, In, and Ag are -0.560, -0.345, and 0.7996 V, respectively, Ga with the most negative electrode potential would act as the cathode, which donated electrons following the reaction $\text{Ga} - 3\text{e}^- \rightarrow \text{Ga}^{3+}$. Then the electrons were transferred from EGaIn LM to the Ag_xIn_y IMC layer of surface distributed FPA nanoparticles, and accepted by the H⁺ in acidic solution to generate hydrogen gas following the reaction $2\text{H}^+ + 2\text{e}^- \rightarrow \text{H}_2 \uparrow$. All the reactions occurred on the surface of magnetic EGaIn LM composite that contacted with acid, and the net reaction could be expressed as $2\text{Ga} + 6\text{H}^+ \rightarrow 2\text{Ga}^{3+} + 3\text{H}_2 \uparrow$. During this process, LM was consumed to produce Ga³⁺, H⁺ was reduced to H₂, while Ag_xIn_y IMC layer simply transferred electrons and remained intact. Therefore, acid did not need to penetrate FPA nanoparticles to cause bubble formation, and would not physically damage the Ag layer. A very small portion of magnetic agents would be released only if their surrounding EGaIn LM was exhausted. So, in the Movie S5, we believed the magnetic EGaIn LM composite was releasing bubbles rather than powders.

- Moreover, the authors mentioned that they reduced the concentration of HCl after describing the above scenario. This implies that controlling the acid's concentration is necessary, indicating that their composite is not stable in acid. However, they do provide evidence that the composite could be relatively stable in the acid found in the stomach.

Response: Thanks for your comment. We totally agree that magnetic EGaIn LM composite will become unstable in highly acidic condition, and controlling the concentration of acid is necessary. We have immersed magnetic EGaIn LM composite in HCl solution with different acid concentrations. As shown in **Fig. R2.2A**, when the concentration of HCl solution was 1 M, EGaIn LM was corroded, accompanied by the constant production of hydrogen bubbles on the surface. However, with the decrease of HCl concentration, magnetic EGaIn LM composite became increasingly stable, especially when HCl concentration was lower than 0.2 M. It was also reported elsewhere that when immersing LM in acidic solution, a HCl concentration lower than 0.2 M was insufficient to thoroughly dissolve the surface oxide film of LM [*Phys. Fluids* 2012, 24, 063101] [*ACS Appl. Mater. Interfaces* 2019, 11, 8685], leading to the disappearance of bubbles.

Considering that the pH of human stomach acid was generally in the range of ~0.9-1.8, HCl solution with a concentration of ~0.1 M was usually employed to simulate gastric

acid. So, combined with **Fig. R2.2A**, no obvious bubbles would be generated if the magnetic EGaIn LM composite was employed in stomach acid. We further incubated 500 mg magnetic EGaIn LM composite in 1 mL 0.1 M HCl solution for different periods, and measured the leakage of metal elements from the composite. As shown in **Fig. R2.2B**, although all metals were found to leak from composite as time went on, the concentration of leaked Ga, In, Fe and Ag was only 17.40, 1.90, 33.52, and 3.94 mg/L, respectively, after 1 h (long enough for targeted transportation application). Considering the total volume of acidic solution was 1 mL, only $\sim 17.4 \mu\text{g}$ Ga and $\sim 1.9 \mu\text{g}$ In, $\sim 33.52 \mu\text{g}$ Fe, $\sim 3.94 \mu\text{g}$ Ag could be dissolved from 500 mg magnetic EGaIn LM composite after 1 h (i.e., only $\sim 0.01 \text{ wt}\%$ of the composite was dissolved), indicating the composite was consumed very slowly.

Fig. R2.2. Stability of magnetic EGaIn LM composite in acidic solution. (A) Optical images of magnetic EGaIn LM composite in HCl solution with different concentrations. (B) The concentrations of leaked metal elements measured in the supernatant of 0.1 M HCl solution containing magnetic EGaIn LM composite over time. Error bars indicate the SD for $n = 3$.

Therefore, combined with the aforementioned experimental results, our magnetic EGaIn LM composite exhibited good stability in the acidic environment of stomach, which fully satisfied the application demands of magnetic miniature soft robot for targeted cargo transportation.

Response 3. Response is acceptable.

Response: Thank you very much. We hope our response can address your concerns at a satisfactory level.

Response to Reviewer #4

The authors have addressed all of my concerns. It should be accepted for publication.

Response: We are pleased that our revisions have satisfactorily addressed your concerns. Thank you very much for your valuable comments, which help us to refine our work and enhance the quality of manuscript.

REVIEWERS' COMMENTS

Reviewer #2 (Remarks to the Author):

The authors responded to my prior comments sincerely, and I find their additional response to Comment No. 2 acceptable.

Response to Reviewer #2

The authors responded to my prior comments sincerely, and I find their additional response to Comment No. 2 acceptable.

Response: We are pleased that our revisions have satisfactorily addressed your concerns. Thank you very much for your valuable comments, which help us to refine our work and enhance the quality of manuscript.